# Data-mining unveils structure–property–activity correlation of viral infectivity enhancing self-assembling peptides

Kübra Kaygisiz [1], Lena Rauch-Wirth [2], Arghya Dutta [3,5], Xiaoqing Yu[4], Yuki Nagata [4], Tristan Bereau[3,6], Jan Münch [2], Christopher V. Synatschke [1] & Tanja Weil [1]

Gene therapy via retroviral vectors holds great promise for treating a variety of serious diseases. It requires the use of additives to boost infectivity. Amyloid-like peptide nanofibers (PNFs) were shown to efficiently enhance retroviral gene transfer. However, the underlying mode of action of these peptides remains largely unknown. Data-mining is an efficient method to systematically study structure–function relationship and unveil patterns in a database. This data-mining study elucidates the multi-scale structure–property–activity relationship of transduction enhancing peptides for retroviral gene transfer. In contrast to previous reports, we find that not the amyloid fibrils themselves, but rather μm-sized β-sheet rich aggregates enhance infectivity. Specifically, microscopic aggregation of β-sheet rich amyloid structures with a hydrophobic surface pattern and positive surface charge are identified as key material properties. We validate the reliability of the amphiphilic sequence pattern and the general applicability of the key properties by rationally creating new active sequences and identifying short amyloidal peptides from various pathogenic and functional origin. Data-mining—even for small datasets —enables the development of new efficient retroviral transduction enhancers and provides important insights into the diverse bioactivity of the functional material class of amyloids.

The stable and safe integration of genetic information is an emerging challenge in cell engineering, e.g., for gene therapy. Retroviral gene transfer is the commonly used method to integrate genetic payloads into cells. Retroviral vectors can deliver therapeutic transgenes with low immunogenicity and hold great promise for the treatment of genetic disorders in recent ex vivo clinical trials[1] and have been approved for chimeric antigen receptor (CAR) T-cell therapy[2].

However, low transduction efficiencies of the viral vectors at target cells resulting from low virus–cell attachment rates are still a major challenge that limits therapeutic efficacy[3,4].

Various materials such as polymers, lipids, and peptides have been introduced as additives to increase viral transduction efficiency mainly by facilitating colocalization of viruses and cells[5]. Interestingly, the transduction-enhancing ability of amyloid peptide nanofibers

[1]Department Synthesis of Macromolecules, Max Planck Institute for Polymer Research, Ackermannweg 10, 55128 Mainz, Germany. [2]Institute of Molecular Virology, Ulm University Medical Center, Meyerhofstraße 1, 89081 Ulm, Germany. [3]Department Polymer Theory, Max Planck Institute for Polymer Research, Ackermannweg 10, 55128 Mainz, Germany. [4]Department Molecular Spectroscopy, Max Planck Institute for Polymer Research, Ackermannweg 10, 55128 Mainz, Germany. [5]Present address: Institute of Biochemistry II, Faculty of Medicine, Goethe University, Theodor–Stern–Kai 7, 60590 Frankfurt, Germany. [6]Present address: Institute for Theoretical Physics, Heidelberg University, Philosophenweg 19, 69120 Heidelberg, Germany. ✉e-mail: synatschke@mpip-mainz.mpg.de; weil@mpip-mainz.mpg.de

(PNF) was first found in seminal fluid in the context of human immunodeficiency virus 1 (HIV-1) infection[6]. This finding resulted in intense research on peptide fibrils to boost viral infectivity[7–11]. For example, the 12-mer peptide enhancing factor C (EF-C, QCKIKQIINMWQ) corresponding to residues 417–428 of the HIV-1 envelope glycoprotein 120 (gp120) was found in a random screening approach and spontaneously self-assembled into positively charged amyloid fibrils, which strongly enhance retroviral transduction[12]. Due to their versatile material properties, amyloid nanostructures have attract broad interest for various therapeutic applications, such as gene therapy[12] or as additives in vaccines[13].

Many peptides and proteins have a propensity to assemble into hierarchically ordered amyloid structures and Nature uses these structures in both, pathogenic and functional contexts[14,15]. A hypothesis that distinguishes pathogenic from functional amyloids is based on the kinetics of their assembly process. It has been proposed that oligomeric species rather than mature, thermodynamically stable amyloid fibrils are related to cytotoxicity[16–18]. Oligomers that prevail for a long time in solution can interact and also refold other monomeric proteins thus acting as multipliers for proteins misfolding and aggregation[19,20]. In contrast, reports on functional amyloids typically show very fast and quantitative assembly without long lag times, thus preventing the formation of cytotoxic oligomers[21,22]. The tensile strength and stability of amyloid fibrils are attributed to their high cross β-sheet structures, which are mainly stabilized by hydrogen bonding and hydrophobic interactions[23]. The defined molecular structure of the peptide building blocks enables modifications of the sequence at the level of single amino acids, which gives access to tuneable assembly features and materials properties from the nano- to the microscale (structure–property), resulting in programmable bioactivity (property–activity)[24].

The observation that certain functional amyloids are biocompatible and exhibit low cytotoxicity combined with virus-binding properties makes them a promising class of materials for retro- and lentiviral gene therapy[12,25,26]. In previous studies, we and others have shown via electron and fluorescence microscopy that virus-particles associate to fibrillar structures[12,26,27]. However, a fundamental understanding of the corresponding molecular as well as nanoscopic property–activity relationship is limited by inherent multi-scale and multi-parameter complexity of the supramolecular self-assembling systems. Data-mining and regression models are efficient tools to study structure–function relationship and unveil underlying design principles and patterns in a database[28,29]. However, so far, the relationship between properties and bioactivity of peptides has been elucidated mainly for antimicrobial[30] or antiviral peptides[31], while they are still elusive for self-assembling nanostructured peptides with transduction enhancing properties.

We have designed and synthesized a peptide library consisting of 163 sequences to study structure–property–activity relationships for infectivity enhancement of nanostructures composed of short peptides. Based on the known and commercially available self-assembling 12-mer peptide EF-C (Fig. 1A)[12], a smaller library was reported by us containing optimized peptide sequences with improved bioactivity[26,32]. However, a fundamental understanding of the structure–property–activity relationship of amyloid peptides and their unique infectivity is still lacking. Therefore, we expand the peptide library by scrambling and substituting amino acids, point mutations, peptide sequence truncations, extensions with bioactive epitopes, and random peptide sequences (Table S2). The resulting peptide library includes peptides with sequence lengths ranging from 4 to 19 amino acids (Table S3, Fig. 1B), whose bioactivity, aggregation, and nanostructure formation were characterized in a quantitative HIV-1 infection assay as well as by transmission electron microscopy (TEM), light scattering count rate, zeta-potential measurements, and

Thioflavin T (ThT) fluorescence assays. Collecting this multi-parameter dataset provided the physicochemical properties for morphology, microscopic aggregation, surface charges, and β-sheet amyloid structures, respectively. Together with bioinformatic parameters, these acquired data were applied in a property–activity correlation analysis using multiple linear regression. Microscopic aggregation of already formed amyloid structures with positive surface charges was identified as the key material characteristics for boosting viral transduction. We demonstrated the general understanding and universal applicability of the key properties for pathogenic and functional amyloids, unrelated to EF-C. To the best of our knowledge, this is the first structure–property–activity relationship of infectivity-enhancing amyloid peptide nanomaterials, that enables not only an understanding but also the rational design and discovery of therapeutic and pathogenic peptide sequences. We envision that the presented study provides a fundamental understanding of this important structural class.

## Results

### Characterization of a peptide library based on EF-C derivatives

To unveil the structure–property–activity relationship of infectivity enhancing peptides, we performed a data-mining approach. This procedure involves data-generation by creating a peptide library, processing and analyses of the peptide measurements, establishing a model for the structure–property–activity relationship, and finally interpretation and prediction based on the established relationship. The peptide library was synthesized via routine solid phase peptide synthesis according to Fmoc-protocol[33] and their purities were confirmed by liquid chromatography mass spectrometry (LC-MS, Fig. S35). Unless stated otherwise, the sequences were obtained in >95% purity by LC-MS from commercial suppliers (Table S3). To induce fibril formation, the peptides were first dissolved in dimethyl sulfoxide (DMSO) and then diluted in phosphate-buffered saline (PBS). The physicochemical properties of the peptides were analyzed on multiple length scales (Fig. 1C). On nanometer scale, the morphologies of molecular assemblies were investigated by TEM and were categorized in (1) fibrillar and (2) non-fibrillar structures (SI Section 1.1, Fig. S1, Fig. S33). Fibril formation was further assessed by the amyloid sensitive ThT dye. Enhancement of fluorescence intensity was accounted to ThT binding to β-sheet rich fibrils (SI Section 1.1)[34]. Both fibril formation and ThT-activity of peptides were evaluated as binary features (fibrils/no fibrils and ThT-active/not ThT-active, Table S5). β-Sheet content was determined via Fourier transform infrared spectroscopy (FT-IR) analysis (Fig. S34) and efficiency to assemble into nanostructures was determined via the conversion rate (CR, Table S5) assay as reported earlier[26]. On the micrometer scale, zeta-potential measurements were conducted to gain information on surface charge of peptide assemblies. Another parameter taken from the zeta-potential measurements was the derived count rate of scattered light, which gives information on turbidity of the sample due to aggregation of fibrillar assemblies (SI Section 1.1, Table S5)[35]. The microscopic aggregation of PNF was further characterized for selected examples via brightfield and fluorescence microscopy (Fig. S32). In this study, we use the term microscopic aggregation to describe the fiber-fiber interaction into less ordered micrometer-sized particles and exclude the highly ordered μm-sized crystalline alignments that have been previously studied for amyloid aggregation[36–38]. Bioinformatic parameters for the library are calculated with amino acid properties to yield global peptide properties like hydrophobicity or isoelectric point (SI Section 1.2, Table S4)[39,40].

As an indicator of biological activity, the transduction enhancement of the self-assembled peptides was assessed by determining HIV-1 infection rates of TZM–bl cells (Fig. 1D). This cell type is a widely applied model cell line for high throughput analysis

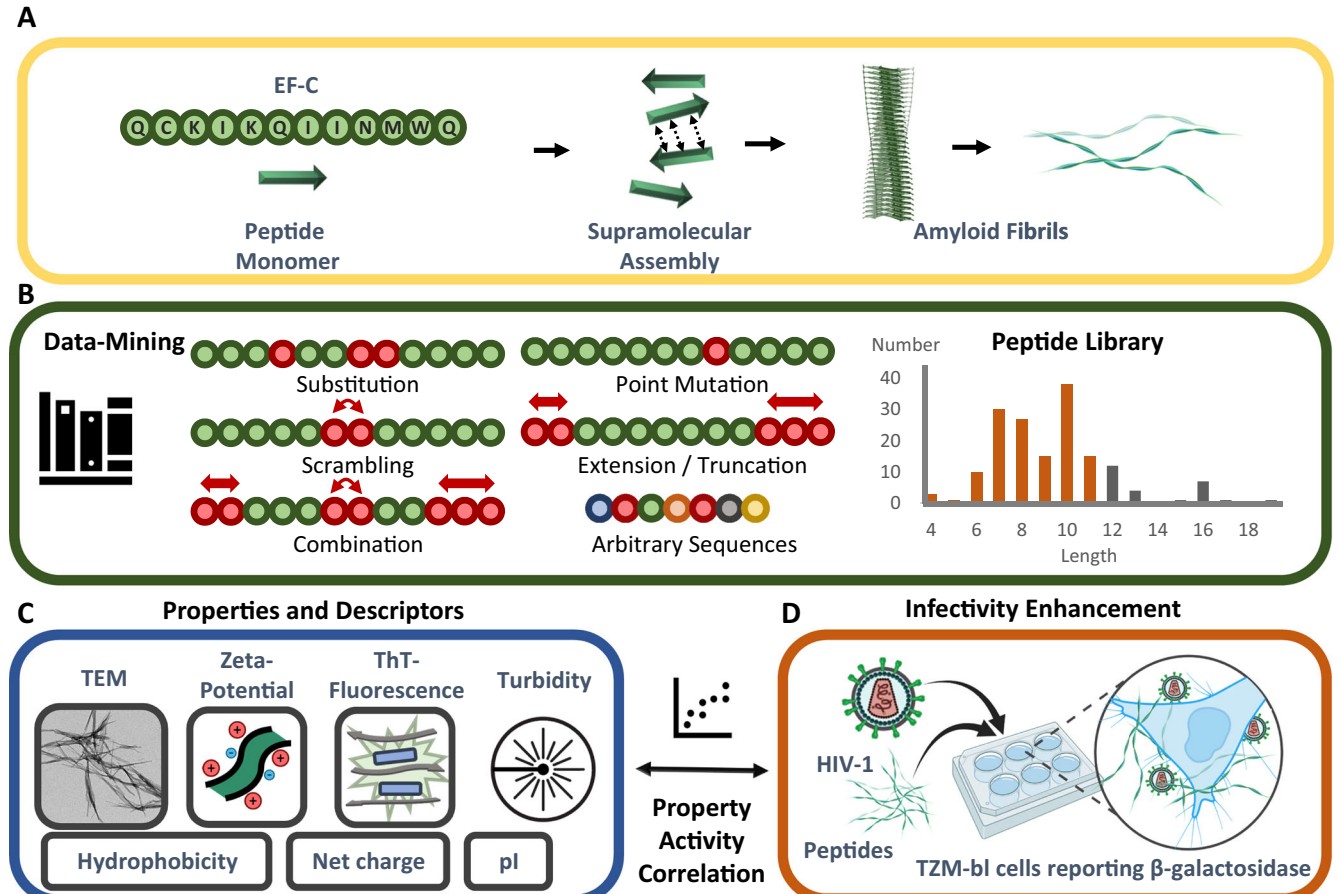

**Fig. 1 | Schematic visualization of this work to identify property-activity relationships of peptides enhancing retroviral transduction enhancement. A** EF-C is a 12-mer peptide sequence, derived from the glycoprotein gp120 of the human immunodeficiency virus 1 (HIV-1), that can assemble via supramolecular interactions into amyloidal fibrils with high cross-β sheet content. **B** Amino acids of the sequence of EF-C were systematically substituted, point mutated, or scrambled. Sequence extensions and truncations have been conducted by repeating terminal amino acids or introducing bioactive peptides. Randomly generated peptides were added to the library with no connection to EF-C. The resulting library consists of 163 peptides, mostly comprising fewer than 12 residues (Table S3). **C** Experimental methods, such as TEM (Fig. S33), zeta-potential, ThT fluorescence (Table S5), turbidity analysis (Table S8) as well as bioinformatic characterization such as hydrophobicity, net charge and isoelectric point (pI) (Table S4) were applied to correlate with (**D**) the infectivity enhancement of HIV-1 on TZM-bl cells by the peptides, as quantified via a chemiluminescence based assay (Table S5). HIV-1 and peptides were incubated together before they were added to cells. Created with BioRender.com. Source data for **B** is provided within the Source Data file.

of HIV-1 infection, because it expresses the HIV-1 receptors CD4 and CCR5 and reports β-galactosidase upon successful infection[41]. HIV-1 was mixed with different concentrations of peptides and subsequently used to infect TZM-bl cells. The infection rates were assessed with a chemiluminescence-based assay, which allowed quantification of the infection rates relative to virus-only infectivity. For example, a 1.3 μM concentration of EF-C fibrils enhanced HIV-1 infection by a factor of 42 ± 7, as compared to infection rates of the virus only control (Table S5). In this study, EF-C at 1.3 μM concentration was selected as a reference, to compare quantitative infection data obtained for every peptide among various measurements (Table S5). Additionally, the cell viability was determined for a representative selection of 77 peptides at a concentration of 1.3 μM and revealed no toxicity (Figs. S2, S3).

The herein investigated peptide library based on EF-C derivatives consists of peptides, which are similar in sequence (Table S2), but show a wide variety of biological activity (*i.e.* infectivity enhancement), which allow to study structure–property–activity relationships. Peptides showing an *n*−fold infectivity enhancement of <4−fold relative to virus-only infectivity or <10% relative to EF-C are considered as inactive, which is the case for 49% of the peptide library (Table S5).

## Microscopically aggregated amyloid fibrils and positive surface charge highly correlate with transduction enhancement

To elucidate the structure–property–activity relationship, we applied linear regression on the experimental and bioinformatic descriptors and evaluated the Pearson correlation coefficient (Pearson *R*). Notably, for most of the descriptors, the logarithm of the infection enhancement relative to EF-C (Log$_{10}$ Infection Rel. EF-C) correlates with a higher Pearson *R* compared to absolute infection enhancement values. To enable linear regression, we therefore applied logarithmic scale for the infectivity relative to EF-C for all the following correlations.

Fibril formation (Fig. 2A, Fig. S5 *R* = 0.59), ThT−fluorescence (Fig. 2B, Fig. S5 *R* = 0.40), and hydrophobicity according to Fauchère hydropathy scale (Fig. S5, *R* = 0.40) correlate strongly with infectivity enhancement. Our data indicate that a high hydrophobicity is important to achieve quantitative assembly of peptides into fibrils (Fig. S6). The importance of the latter is further demonstrated with a representative selection of 80 peptides (Table S5), which we additionally characterized regarding the conversion of monomers to nanostructures (Fig. S5, conversion rate, *R* = 0.50, Fig. S7 B, Table S5).

Previous reports indicated that the zeta-potential representing the surface charge of the fibrils plays a crucial role for transduction enhancement[6,12,25,27]. Plotting the zeta-potential against Log$_{10}$ Infection

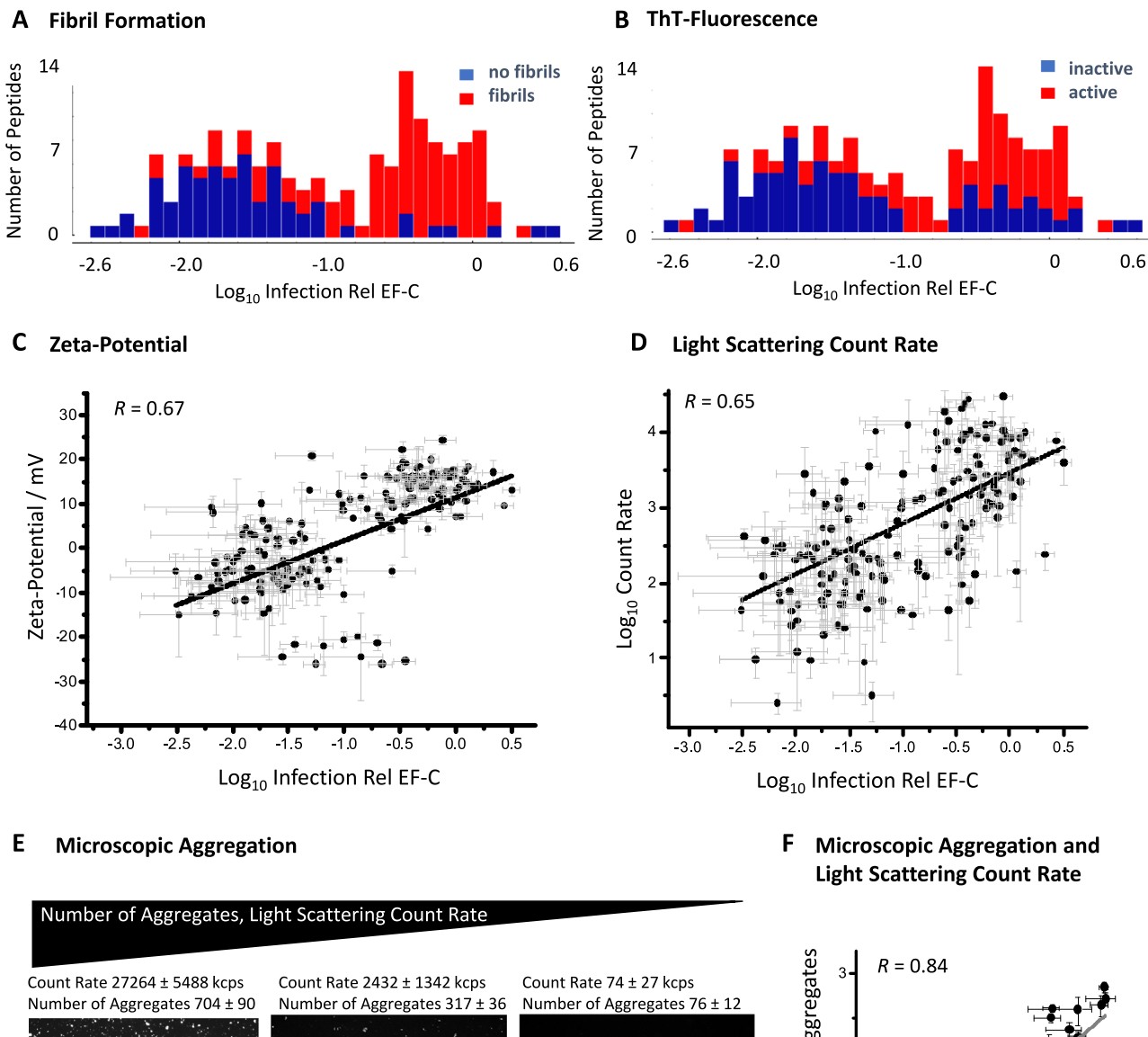

**Fig. 2 | Correlation of experimental parameters with $Log_{10}$ Infection Rel EF-C of a library containing 163 peptides. A** Histogram of $Log_{10}$ Infection Rel EF-C showing fibril forming (red) and non-fibril forming (blue) peptide distribution analyzed via TEM (Fig. S33). **B** Histogram of $Log_{10}$ Infection Rel EF-C showing ThT-active (red) and ThT-inactive (blue) peptide distribution (Table S5). **C** Zeta-potential plotted against $Log_{10}$ Infection Rel EF-C. Mean values are displayed with error bars indicating standard deviation from triplicate measurements (Table S5). Linear fit with Pearson correlation factor $R = 0.67$. **D** Count rate of scattered light plotted against $Log_{10}$ Infection Rel EF-C. Linear fit with Pearson correlation factor $R = 0.65$. Mean values are displayed with error bars indicating standard deviation from triplicate measurements (Table S5). **E** Selection of peptides with high and low count rate showing relationship between number of particles and count rate of scattered light (Table S8). Peptide fibrils were preincubated at 1 mg/mL one day and diluted to 0.1 mg/mL with ThT solution (50 μM) just before measurement (Fig. S32). Number of aggregated fibrils ( > 10 μm²) were quantified via fluorescence microscopy in an area of 1330 μm × 1330 μm in triplicate measurements via automated counting (3D object counter ImageJ), scale bar 200 μm. **F** $Log_{10}$ Count rate from zeta-potential measurements of representatively selected peptides plotted against averaged number of aggregated particles in an area of 1330 μm × 1330 μm (Table S8). Mean values are displayed with error bars indicating standard deviation from triplicate measurements, linear fit (solid line) demonstrates a high correlation with a Pearson $R$ of 0.84. Source data for **A**–**D** and **F** is provided within the Source Data file.

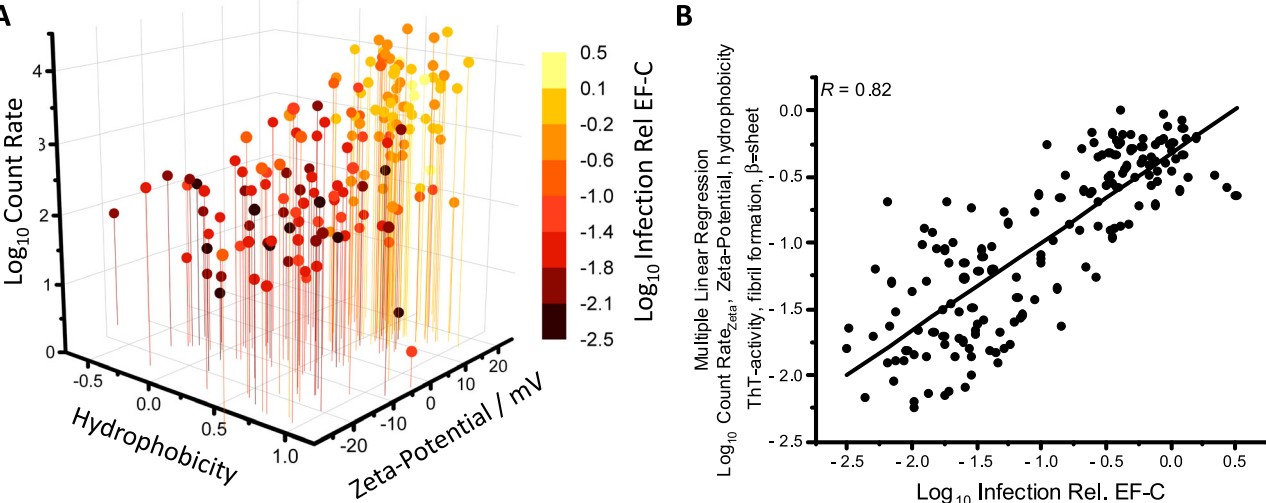

**Fig. 3 | Multiple parameter analysis to evaluate property-activity relationship between physicochemical and bioinformatic descriptors and infectivity enhancement. A** Four-dimensional plot visualizes that a positive zeta-potential (Table S5), high hydrophobicity (Table S4), and high count rate of scattered light (Table S5) are corresponding to infectivity enhancement (color coded $Log_{10}$ Infection Rel. EF-C, Table S5). **B** Multiple linear regression (constrained linear least-squares problems solution) with the equation: $y = A0 + A1*x1 + A2*x2 + [...] + A5 * x5$ with best performing single parameters of 163 peptides based on EF-C

(Fig. S5, Table S4, S5): $Log_{10}$ Count Rate, zeta-potential, hydrophobicity, ThT-activity, fibril formation, β-sheet and $Log_{10}$ Infection Rel EF-C as dependent target variable show an improved linear correlation by 22% compared to the best correlating feature zeta-potential. Linear Fit with Pearson correlation factor $R$ is 0.82. The found relationship is $Log_{10}$ Infect Rel Ef-C = −2.33462 + 0.02128 (zeta-potential) + 0.29879 ($Log_{10}$ Count Rate) + 0.27355 (fibril formation) + 0.26241 (hydrophobicity) + 0.10744 (ThT-activity) + 0.00356 (β-sheet). Source data for Fig. 3 A, B is provided within the Source Data file.

relative to EF−C shows indeed the strongest linear correlation ($R = 0.67$, Fig. 2C, Fig. S5, Table S5). Another comparably high correlating property is microscopic aggregation of the peptides, which was measured via the count rate of scattered light during zeta-potential measurement ($Log_{10}$ Count Rate, $R = 0.65$ Fig. 2D, Fig. S5, Table S5, Table S8). The timescale for the aggregation of peptides is within seconds as can be observed via TEM and turbidity from light scattering (Fig. S4). Although there is no suitable model for correlating the scattering intensity of light with the size of PNFs (SI Section 1.1), our data shows that the logarithmic number of μm−sized aggregated fibrils determined by fluorescence microscopy (Fig. 2E, Fig. S32) correlates linearly with the logarithmic scattering intensity (Fig. 2F, Table S8), which further validates the relationship between the microscopic aggregation and infectivity.

Despite the high correlation of zeta-potential and count rate with infectivity enhancement (Fig. 2C, D, Table S5), some data points deviate from the regression line. Moreover, not all bioactive peptides also form fibrils−for example, 7 infectivity-enhancing peptides from the library (Fig. 2A) form β-sheet rich amorphous aggregates instead. Interestingly, these peptides show a high calculated hydrophobicity due to large amounts of tryptophan (W) and tyrosine (Y) residues (Fig. S16), which indicates that multiple parameters are codependent of each other and cannot be covered from single parameter models to describe the property–activity relationship sufficiently.

### Synergistic effect of hydrophobicity, charge, and microscopic aggregation

We hypothesized that several conditions must be fulfilled to observe peptide−virus−cell interactions, which ultimately leads to enhanced viral transduction. So, in a next step, we considered multiple descriptors at once to identify the relevant physicochemical properties. Plotting the best correlating continuous features (zeta-potential, count rate, hydrophobicity) against $Log_{10}$ infection relative to EF-C directly visualizes that a positive zeta-potential, high count rate and high hydrophobicity are coinciding with infectivity enhancement (Fig. 3A, Table S5).

In fact, applying multiple linear regression on the best correlating features revealed a superior Pearson correlation parameter ($R = 0.82$) compared to the single features alone, an increase by 22% compared to best correlating single feature zeta-potential ($R = 0.67$) (Fig. 3B, Table S5). A strong effect of multi-parameter considerations is observed for the β−sheet content, which was previously identified as a critical feature for infectivity in a small data set[26]. In our extended peptide library, we observe that the presence of β-sheet rich structures is crucial for activity only if we apply multi-parameter regression ($R = 0.91$, Fig. S10D, E) but not from single-parameter correlations ($R = 0.29$, Fig. S5, Table S5). This is because peptides with similar β−sheet content are not enhancing transduction efficiency to the same extent (Fig. S7) because they either do not show positive surface charge, do not form fibrils or do not aggregate (Fig. S10A−C).

Applying multi-parameter regression has its strongest impact on correlation of the bioinformatic parameters. For example, the highest single-parameter bioinformatic correlation can be found for hydrophobicity ($H_{fauchere}$, $R = 0.40$, Fig. S5). However, in combination with all other calculated descriptors the correlation coefficient increases by roughly 70% (Fig. S11, $R = 0.67$, Table S4). The advantage in applying multiple linear regression is, that each contribution is weighted and reveals the importance of combined properties for the activity. In this way a statistically significant ($p < 0.05$) correlation between zeta-potential, count rate, fibril formation, and infectivity enhancement (Fig. S10D) for the experimental parameters and hydrophobicity ($H_{fauchere}$), aliphatic index (AI), hydrophobic moment index (HMI), instability index (II), graph shape index (GSI), upsilon steric parameter (USP), polarizability (P) and normalized van der Waals Volume (Norm vdW Vol.) (Fig. S11) for the bioinformatic parameters are identified. We propose that next to hydrophobicity, also other global peptide features (P, AI, HMI, USP, Norm. vdW Vol., GSI and II) correlate significantly with infectivity enhancement ($p < 0.05$) but with a comparably small correlation coefficient (Figs. S5 S10, and S11). These parameters calculate different aspects of the amino acid side chain hydrophobicity, charge, sterical demand, and the sequence amphiphilicity (SI Section 1.2).

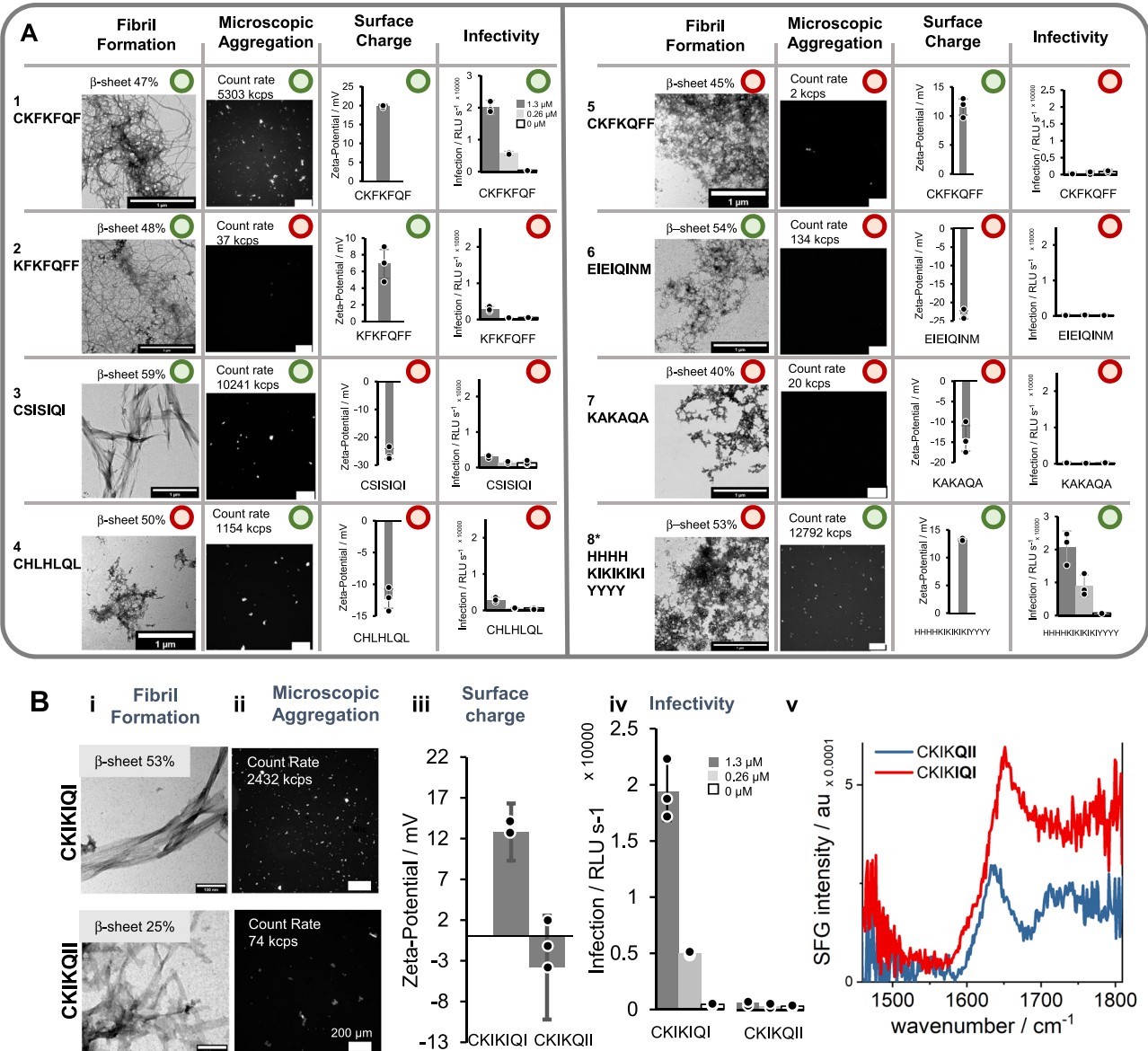

**Fig. 4 | Evaluation of global property–activity relationship for infectivity enhancement with selected peptides as case studies to gain insight into the mode of action.** Internal order is evaluated by TEM (Fig. S33, Table S5, fibril formation) and FT-IR (Fig. S34, Table S5 β-sheet content), surface charge by zeta-potential and infectivity is shown in absolute infection rates of peptides at 1.3, 0.26 and 0 μM (Table S5). All displayed error bars indicate standard deviation from triplicate measurements. **A** Overview of all possible combinations of physicochemical features, which correlate most strongly with infectivity enhancement with examples for each possible combination. 1 The majority of peptides (75%), which enhance infectivity have high internal order, aggregate into μm-sized aggregates, and have a positive zeta-potential, as shown exemplarily for the peptide CKFKFQF. 2 Without microscopic aggregation there is no infectivity enhancement even though other prerequisites are fulfilled such as fibril formation and positive zeta-potential shown for KFKFQFF. 3 Peptide fibrils which aggregate to clusters but have a negative surface charge are not enhancing viral infection, shown for CSISIQI.

However, if the peptides are highly hydrophobic slight infection enhancement can be observed (e.g. for EIEIQINMWQ, Fig. S15). Peptides which do not fulfill at least two of the features (assembly, aggregation, charge) cannot enhance infectivity as exemplarily shown for 4 (CHLHLQL or CKFKFQF with EGCG, Fig. S21), 5 CKFKQFF 6 EIEIQINM, 7 KAKAQA. 8* Peptides with a high amount of hydrophobic amino acids (W, Y) can enhance transduction although they are not assembling into fibrils if they form β-sheet rich, positively charged aggregates e.g. shown for HHHHKIKIKIYYYY. **B** Comparison of physicochemical properties and activity of CKIKIQI and CKIKQII. i TEM micrographs and β-sheet content determined by FT-IR (Fig. S34), scale bar 1 μm. ii Fluorescence microscopy of ThT-stained peptide fibrils with their respective light-scattering count rate (Table S5), scale bar 200 μm. iii Zeta-potential iv Absolute infection rates of peptides at 1.3, 0.26 and 0 μM concentration. v SFG spectroscopy at the amide I and II region. Source data for **A**, **B** is provided within the Source Data file.

## A differentiated perspective on the mode of action

To showcase the underlying codependences of the internal order (β–sheet and fibril formation), microscopic aggregation, and zeta-potential in the observed property–activity relationship, we exemplarily selected individual peptides (Fig. 4A, B, SI Section 6).

The most prevalent case (38% of the library, 75% of all active peptides, Table S5) for infectivity enhancement are peptides

assembling into fibrils with β-sheet secondary structure, which further aggregate into positively charged μm–sized aggregates (for example CKFKFQF, Fig. 4A-1).

The importance of these parameters is shown by the lack of activity when removing one of the properties (Fig. 4A). For example, if peptides assemble into positively charged fibrils but do not aggregate into μm-sized clusters (e.g. KFKFQFF, Fig. 4A-2), no strong infectivity

enhancement can be observed. In our library, cases where an assembly into fibrils but no aggregation into μm−sized particles occur, are mostly observed for peptides, which do not contain N−terminal cysteine at the first position (Fig. S13, SI Section 6.1). For example, the peptides CKIKIQI and KIKIQIC only differ in the position of cysteine within their sequence, but CKIKIQI shows stronger aggregation and infectivity enhancement than KIKIQIC (Fig. S14). In contrast to lysine, cysteine is an amino acid with a hydrophobic residue, and might increase the surface hydrophobicity of the fibrillar structure when positioned at the N−terminus. The thiol group of the cysteine side chain can further form disulfide bonds, which might increase inter-fibril attachment and aggregation.

Small changes in molecular sequence can induce a strong change in microscopic aggregation properties as observed for example for CKIKQII and CKIKIQI. Sum−frequency generation (SFG) measurements can give information on water interaction of fibrils, which is dependent on surface hydration and hydrophobicity properties of fibrils[42,43]. To investigate changes in water−fibril interaction the peptides CKIKIQI and CKIKQII were selected as examples since they are composed of same amino acids in different order, form fibrils and differ significantly in their aggregation behavior (Fig. 4B). The SFG measurements show that while both peptides accumulate at the air/water interface, the signal intensity of CKIKIQI is higher than CKIKQII. This indicates that CKIKIQI is more ordered or may be present at higher concentrations at the air/water interface (Fig. 4B). Both effects can be traced back to a higher surface hydrophobicity and lateral interaction of fibrils in μm−sized aggregates, which in turn are a prerequisite for infectivity enhancement.

Noteworthy, not every parameter in this relationship must be fulfilled to observe any infectivity enhancement. For example, while the majority (90%) of infectivity-enhancing peptides (Infection relative to EF-C > 10%) forms fibrils, some of the active peptides show non−fibrillar morphologies (Fig. 2A). These peptides contain a high amount (> 20%) of hydrophobic amino acids such as W or Y and form amorphous amyloid aggregates with high β−sheet content (Table S5). Additionally, they fulfill other prerequisites such as positive zeta-potential and microscopic aggregation (Fig. S16). Fibril formation is strongly connected to aggregation as it likely facilitates entanglement but appears not to be an indispensable prerequisite for infectivity enhancement (Figs. 4A-8, S16, and S19).

As already extracted from the global correlation, the surface charge plays a key role. Only weak infectivity enhancement is observed for peptides with negative zeta-potential even if the fibrils with high β−sheet content are aggregating (Fig. 4A−3, Table S5). However, as it also can be extracted from the single parameter correlation with zeta-potential (Fig. 2A, Table S5), five peptides with negative zeta-potential show a moderate infectivity enhancement larger than 10% Rel EF-C (Log$_{10}$ Infect > −1) (Fig. S15). We hypothesize that these peptide fibrils can slightly enhance infectivity because of their extraordinarily high hydrophobicity (> 0.6) and microscopic aggregation of fibrils, which favor interactions with viruses and cells despite the electrostatic repulsion. The importance of considering hydrophobic interactions in addition to electrostatic attraction is further emphasized by a recent report from us, in which we applied machine learning approach to target different sequence spaces[44]. An overall lower hydrophobicity− be it by substituting phenylalanine (F) or isoleucine (I) amino acids with alanine (A) or leaving out hydrophobic amino acids−results in loss of activity (SI Section 6.3−6.4, Figs. S17−S20), further emphasizing its impact.

Activity is also lost, when more than one of the criterions (fibril formation, positive zeta-potential, aggregation) we identified is not fulfilled (Fig. 4A−4−7). Peptides that only show μm−sized aggregation (Fig. 4A−4), positive zeta-potential (Fig. 4A−5), or β−sheet rich–fibrils (Fig. 4A−6) do not enhance viral infectivity. To illustrate that the

activity of a peptide can be solely traced back to physicochemical properties while the sequence stays same, we added a chaotropic substance (Fig. S21)[45]. Adding epigallocatechin gallate (EGCG) to CKFKFQF drastically decreases infectivity enhancement, β−sheet, and surface charge while microscopic aggregation still takes place (Fig. S21). This once more demonstrates that positive surface charge of microscopically large aggregates with a β−sheet structure are pre-requisites for activity.

In summary, our findings show that higher hydrophobicity of assembled peptides drives microscopic aggregation, which is also a prerequisite for virus−cell membrane interaction and colocalization additional to electrostatic attraction. Thereby, we conclude that mul-tiple conditions including (I) a positive surface charge, (II) ordered structure, (III) microscopic aggregation are required for highly effi-cient retroviral transduction enhancement (Fig. 5).

## Design rules: alternating, amphiphilic sequences patterns

We wondered if a sequence pattern exists, that underlies the identi-fied key physicochemical properties and results in infectivity enhancement. To find common design rules which are preserved in active peptides, the sequences in the library were simplified by coarse−graining the amino acids according to their side chain hydropathy (Hydrophilic P, Hydrophobic H) and charge (−, 0, +). The distribution of recurring patterns in highly active peptides (Infect. Rel. EF-C > 70%) was counted (SI Section 7, Fig. S22A)[46]. The majority (55%) of the active sequences and 44 peptides in the library (163 peptides, Table S3) were composed of alternating amphiphilic sequences P$^+$H$^0$P$^0$H$^0$P$^0$H$^0$P$^0$ or P$^+$H$^0$P$^+$H$^0$P$^0$H$^0$P$^0$ (Fig. 6A).

Interestingly, an alternating amphiphilic sequence does not guarantee high transduction efficiency: 8 out of 44 peptides have an alternating amphiphilic sequence but do not show transduction enhancement. This is likely due to their low sequence hydrophobicity (Fig. S23A). To systematically study the impact of hydrophobicity on activity, amino acids from highly active sequences (W, M, N, I, C, Q, K, F and Y, Fig. S22B, Fig. 6A, B) and from weakly active peptides (S, G, R, H, T, V, P, A and L, Figs. S22B and S23B)[46] were used to representatively create peptides which share the same amphiphilic pattern (P$^+$H$^0$P$^0$H$^0$P$^0$H$^0$P$^0$ or P$^+$H$^0$P$^+$H$^0$P$^0$H$^0$P$^0$) and test their infectivity enhancement. Noteworthy, we observed that most of the herein-found amino acids from highly active sequences and weakly active sequences have been known to promote and break β-sheet fibril formation, respectively (SI Section 7).

This approach was exceptionally effective in generating new and active sequences: Eight of the 10 newly created peptides composed of the identified pattern and amino acids prevalent in highly active sequences show infectivity enhancement (Fig. 6B, Table S6). Out of these only the W−rich peptide RFWFWFW does not form fibrils (Figs. 6B and S25) but fulfills all other important requirements for infectivity enhancement−especially a high hydrophobicity (Fig. 6B). Reducing the number of W by one decreases hydrophobicity (1.58 → 1.18) and results in loss of the infectivity enhancing properties as shown for (RINFWFW, Fig. 6B and E). A high amount of W is related to more disordered fibrils but increased activity of the peptides−as also observed for other examples such as KIKIWIW → KIWIWIW (Fig. S25) or C−terminal extension with W (Fig. S16 and S20). These examples once more emphasize the codependences of multiple descriptors for infectivity-enhancing peptides and enable the creation of new active sequences in a reliable fashion. Strikingly, in this way a cysteine- and glutamine-rich peptide KCQCQCQ was found, which showed activity in the range of EF-C (Fig. 6B, Table S6) while being half the length of EF-C (Fig. 6B, Table S6).

The active peptides all share the identified key-properties for infectivity enhancement, which emphasizes the importance of an

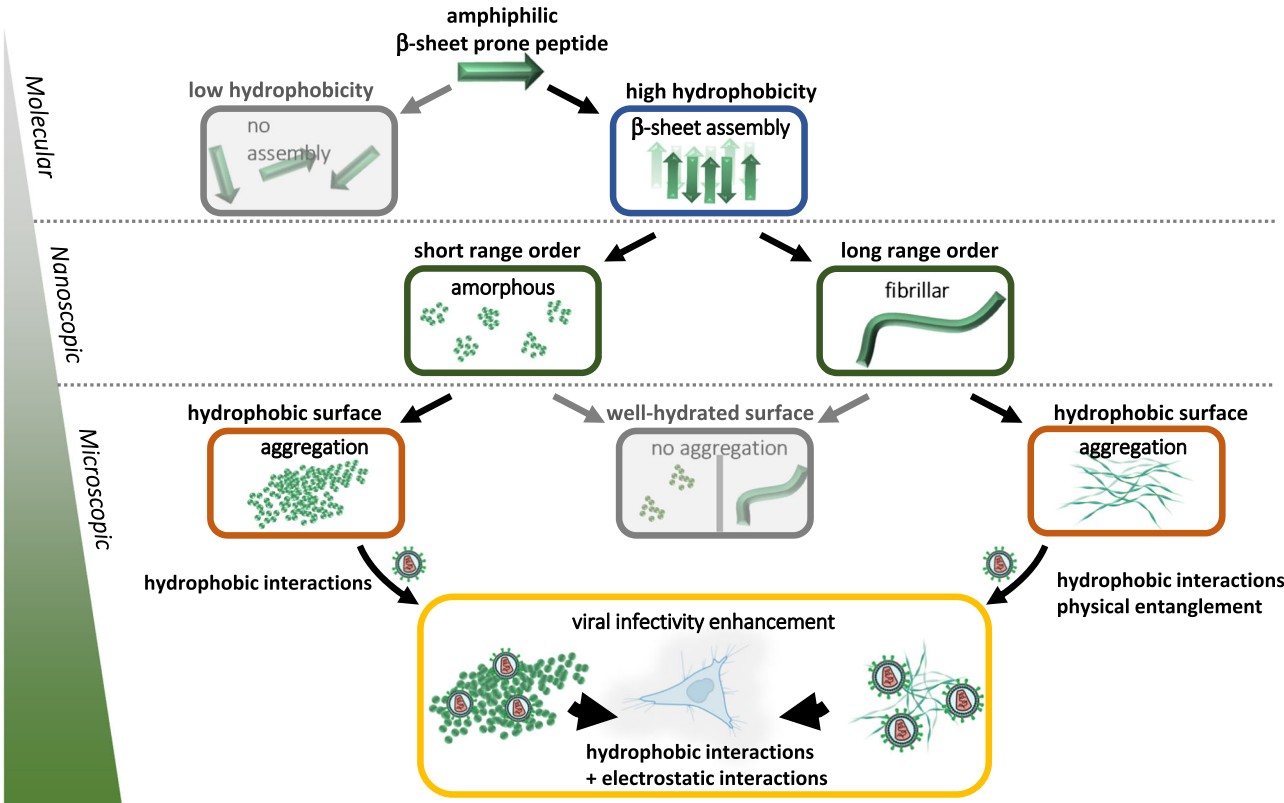

**Fig. 5 | Schematic overview of proposed mode of action over multiple scales from molecular to microscopic level.** Molecular—Amphiphilic peptides with certain hydrophobicity can form β-sheet rich structures. Nanoscopic—Depending on the hydrophobicity of the β-sheet structures, short-range ordered amorphous aggregates or long-range ordered fibrils can form and further aggregate into μm-sized particles. Microscopic—β-sheet rich aggregates can bind viruses and interact with cellular membrane by electrostatic and hydrophobic interactions. Created with BioRender.com.

amphiphilic sequence pattern with sufficiently hydrophobic amino acids and cationic amino acids for aggregation of self–assembled peptide fibrils.

## Key properties for infectivity enhancement translate to functional and pathogenic amyloids

The fact that aggregation is necessary to observe transduction enhancement may shed light on amyloids in pathogenic contexts. In Nature, microscopic aggregation of amyloidal fibrils is well known as so-called amyloid–plaques, which results in deposition of non-soluble proteins in neurodegenerative diseases as well as facilitating adhesion of bacterial biofilms on substrates[14]. Amyloid plaques from amyloid Aβ occur in several neurodegenerative diseases and may have a connection to neuroinflammation processes. It is speculated that amyloid Aβ is produced by the immune system to combat infections[47,48]. In a downstream process, these amyloid fibrils might form micron–sized aggregates (amyloid plaques) that facilitate infections, leading to stronger infectious reactions—which eventually produce more amyloid fibrils in a vicious cycle. We hypothesized that the property–activity findings from the EF-C library can be translated to other amyloid peptides from functional and pathogenic contexts. To test key–properties for infectivity enhancement, we selected amyloidal peptides from protein databases and literature reports based on published data from transmission electron micrographs and ThT-fluorescence assays (Table S1).

Three of the 24 peptides that were selected, namely ISK-LEYSNFSVRY, LANWMCLAKW and VHDCVNITIK, showed infectivity enhancement of retroviral transduction in the range of 30–40% relative to EF-C (Fig. 7A, Table S7). These newly found transduction-enhancing peptides have a positive zeta-potential (Fig. 7B, Table S7, + 9.3 mV, +14.7 mV, +9.3 mV, respectively), microscopically aggregate

(Fig. 7B, Fig. S29-B, Table S7), assemble into fibrils (Figs. S36, S37, and 7C–E) with a high β–sheet amount (Figs. S30 and S39, Table S7) and are ThT-active (Fig. S29–A). We studied the interaction of fibrils with HeLa cells by applying confocal fluorescent laser scanning microscopy. The interaction of the amyloidal fibrils with HeLa cells is clearly visible in Fig. 7C–E and shows that these newly discovered infectivity-enhancing peptides act in a comparable fashion as EF-C by associating with cellular membranes and facilitating viral uptake through colocalization[12].

Interestingly, four fibril-forming peptides IKFLSVN, GYVIIK, RQGNINIVA, and MKVIFLKDVKG, that are derived from peroxir-edoxin III[49], serum amyloid P[50], insects' chorion[51], and ribosomal protein[52], respectively, have positive zeta-potential in a similar range, but do not enhance infectivity. This is because these peptides do not aggregate to microscopic clusters as visible from low light scattering count rates (Figs. 7B and S29–B) and microscopy data (Fig. S31).

In summary, studying endogenous amyloidal peptides confirms the identified key–properties for transduction enhancement and further emphasizes the importance of microscopic fibrillar aggregates beside the electrostatic interactions to mediate binding of the positively charged fibrils to the negatively charged cell surface.

## Discussion

In this study, we aimed to understand the mode of action of short peptides acting as adjuvants to increase viral infectivity. We hypothesized that certain physicochemical properties are responsible for the infection-enhancing properties of peptides. To test this, we applied a data-mining approach that includes the creation of a suitable dataset, the evaluation of corresponding data, and interpretation of found relationships. Therefore, we created a peptide library based on known

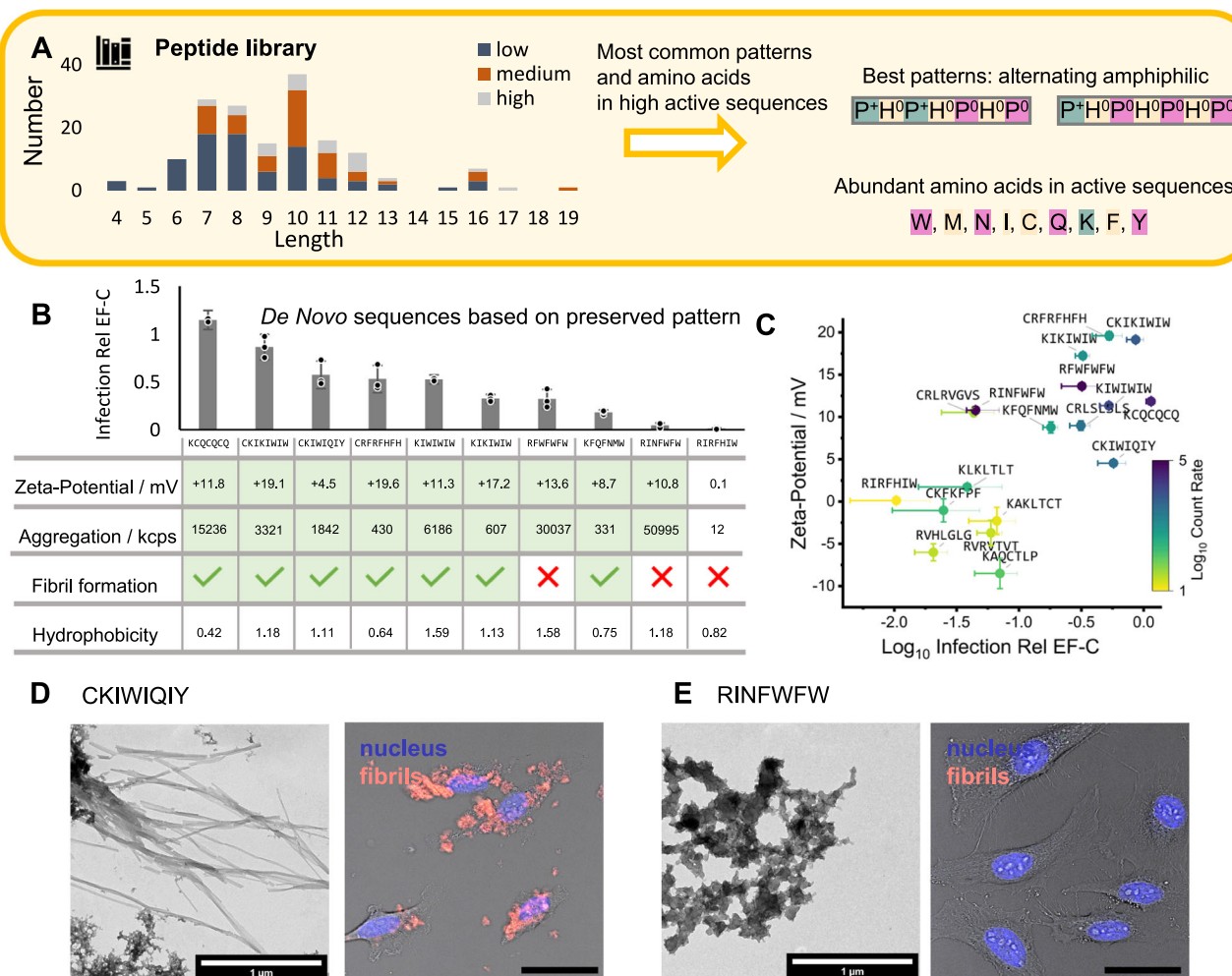

**Fig. 6 | Preserved pattern in infectivity enhancing peptides can be applied to create new active sequences confirming property-activity relationship found for original library. A** Coarse graining infectivity enhancement in low (<10%) medium (10-70%) and high (>70%, Infection Rel EF-C) and coarse graining amino acids according to their hydrophilicity (H = hydrophobic; P = hydrophilic) and side chain charge at pH 7.4 (+ = positive, − = negative, 0 = neutral) enables identification of most abundant sequence pattern in highly active peptides (Table S5) that is an alternating amphiphilic motif $P^+H^0P^+H^0P^0H^0P^0$ and $P^+H^0P^0H^0P^0H^0P^0$. Source code is available under the following link https://doi.org/10.5281/zenodo.8079726[46].
**B** New peptides are created by combining the preserved pattern $P^+H^0P^+H^0P^0H^0P^0$ with the identified best-performing amino acids. 8 of 10 newly created peptides are enhancing infectivity (>10% Infection Rel EF-C at 1.3 μM, Table S6) and overall confirm the identified property-activity relationship. Error bars indicate standard deviation from triplicate measurements. **C** Graphical visualization of the most important property-activity relationship zeta-potential plotted versus Log$_{10}$ Infection relative to EF-C. The color code represents the Log$_{10}$ Count Rate from light scattering, which can be traced back to microscopic aggregation (Table S6). Mean

values are displayed with error bars indicating standard deviation from triplicate measurements. **D** Selected representative example for fibril morphology (TEM, Fig. S25, scale bar 1 μm) and cell interaction (confocal microscopy, scale bar 50 μm) of an active peptide (CKIWIQIY), which is based on the preserved pattern. Representative confocal image shows merged measurement from transmission, Proteostat (red, fibrils) and Hoechst (blue, cell nuclei) channels. **E** Selected representative example for amorphous morphology (TEM, scale bar 1 μm) and no cell interaction (confocal microscopy, scale bar 50 μm) of an inactive peptide (RINFWFW), which is based on the preserved pattern. Confocal image shows merged measurement from transmission, Proteostat (red, fibrils) and Hoechst (blue, cell nuclei) channels. CKIWIQIY and RINFWFW were both susceptible for amyloid sensitive dye staining (Fig. S24), but RINFWFW does not attach to cells in accordance with its low activity. TEM measurements were conducted once with at least three microscopy images recorded for each peptide sample, confocal images have been prepared in biological triplicates and recorded at least at three different sites. Source data for **A**–**C** is provided within the Source Data file.

infectivity-enhancing peptide EF-C. This enables tracing back differences in activity to specific physicochemical properties because sequences stay relatively similar.

Since the interactions between fibrils, viruses and cells occur on multiple length scales we focus on applying complementary methods to characterize multiple physicochemical properties for a large number of peptides, thus circumventing scale limitations of individual characterization methods. For example, TEM is a qualitative high-resolution method for structural characterization on a nanometer scale, but it cannot quantify reliably fibril polymorphism and it lacks information on intermolecular interactions, such as β−sheet content, which was therefore determined via FT−IR spectroscopy. It is

well-known that amyloid polymorphism can greatly influence its bioactivity[53,54]. However, the systematic study of different fibril morphologies is outside the scope of this study. To detect the formation of β-sheet rich fibrils in bulk, we utilized the amyloid-sensitive ThT dye. However, it should be noted that ThT-fluorescence has been associated with false-positive results (SI Section 1.1). All these methods cannot give information on microscopic aggregation properties, for which we applied microscopy and light scattering. Bioinformatic properties gave information for elucidating general effects of structural changes on infectivity enhancement. Multiparameter analyses enabled us to correlate multi-scale dependencies of interaction of materials with cells and viruses.

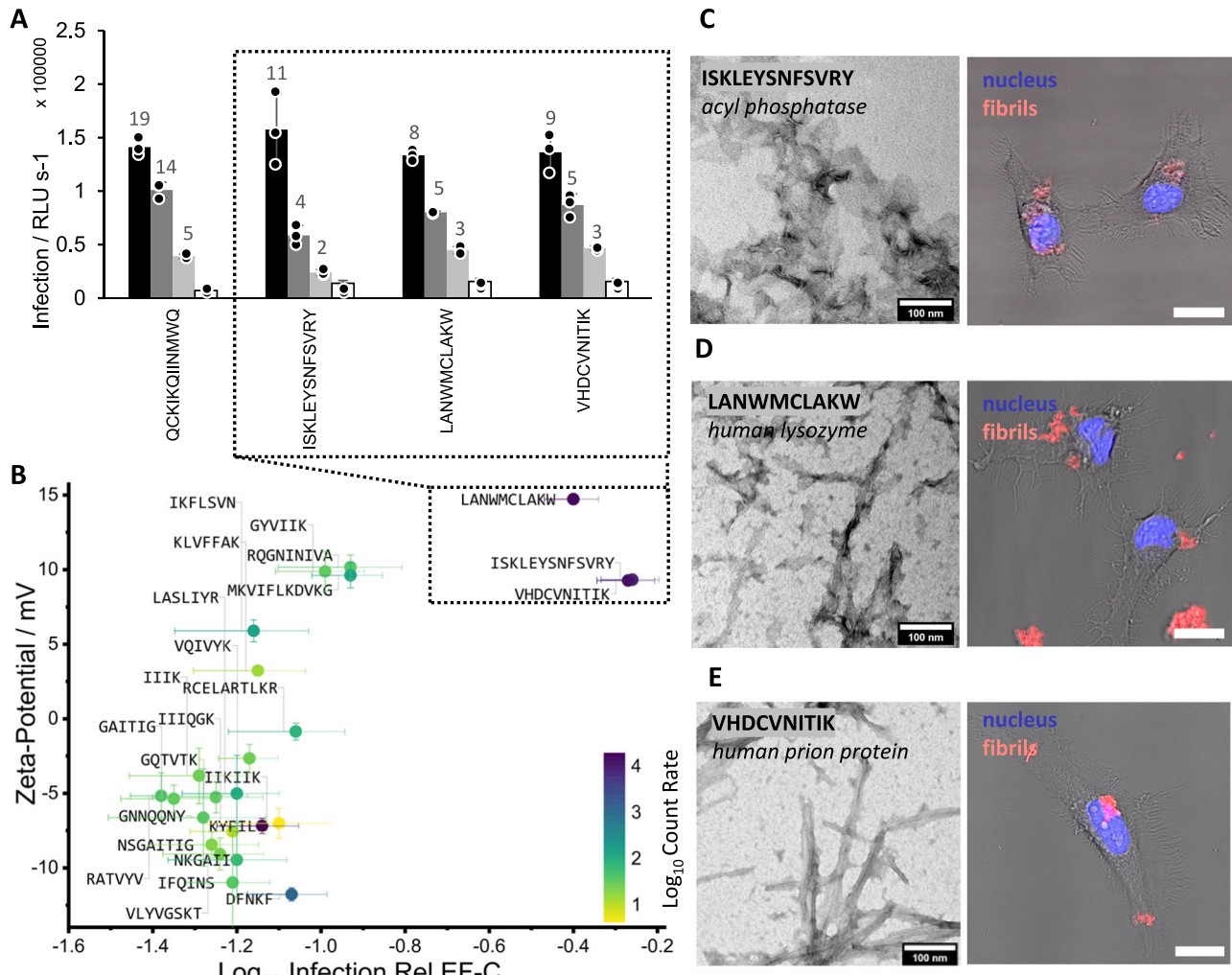

**Fig. 7 | Infectivity enhancing properties of selected amyloid peptides from pathogenic and functional origin. A** Infection rates of EF-C, and peptides ISK-LEYSNFSVRY, LANWMCLAKW, VHDCVNITIK at 6.5, 1.3, 0.26, 0 μM concentration. Numbers at the end of bars indicate n-fold infectivity enhancement compared to virus-only infectivity (0 μM) (Table S7). Error bars indicate standard deviation from triplicate measurements. **B** Zeta-potential plotted against Log₁₀ Infection Rel EF-C. Color indicates Log₁₀ Count Rate by scattered light. Three peptides (box) that show identified important physicochemical properties that have positive zeta-potential, show fibril formation, and aggregation are enhancing infectivity (Table S7). Mean values are displayed with error bars indicating standard deviation from triplicate

measurements. TEM micrograph (1 mg/mL, PBS, Fig. S36, S37 scale bar 100 nm) and confocal microscopy (scale bar 20 μm) of acyl phosphatase derived peptide ISK-LEYSNFSVRY (**C**), human lysozyme derived peptide LANWMCLAKW (**D**) and human prion protein derived peptide VHDCVNITIK (**E**). 20 μg/mL preassembled peptides (stained with Proteostat, red) were incubated with Henrietta Lacks (HeLa) cells (nucleus stained with Hoechst, blue). For single channels see Fig. S28. TEM measurements were conducted once with at least three microscopy images recorded for each peptide sample, confocal images have been prepared in biological triplicates and recorded at least at three different sites. Source data for **A**, **B** is provided within the Source Data file.

We observed that the interactions of peptides with cells and viruses are mostly driven by three codependent properties: (I) ordered structure (II) microscopic aggregation and (III) positive surface charge. While the importance of zeta-potential and fibril formation was already suggested for EF-C PNF in previous reports[12,26,55], microscopic aggregation—which we found to correlate strongly with infectivity—was not systematically studied nor identified as a necessary feature before.

We hypothesize that μm−sized aggregates can accelerate infectivity by physical entanglement with cellular protrusions, which are actively transporting the fibril/virus complexes to the cell surface, where virions can enter the cells[55]. Interestingly, for aggregates larger than 10 μm² it is not the size, but rather the number of aggregates correlating with infectivity enhancement. Fibrils aggregating to few very large aggregates might simply not offer enough particles for virus−cell interactions or may be too big to get efficiently engaged by protrusions. In contrast, fibrils aggregating to many small μm−sized aggregates provide a high number of dispersed particles for viruses to

attach to and subsequently interact with cell membranes. Noteworthy, there are no obvious changes in fibril formation and aggregation properties of the fibrils in the presence of virions, based on our own data and earlier reports[12,25]. The importance of well-distributed aggregates rather than aggregate size is emphasized by findings from cumulative surface area of total aggregates, where we do not observe any difference between the surface area of "active" aggregates and "inactive" aggregates (SI Section 4.3, Fig. S8). Interestingly, the formation of certain aggregate size and number appears to be an intrinsic feature of the non-covalent self-assembly properties of the molecular sequence, *e.g.* after mechanical downsizing aggregates by pressure or ultrasonication, their original size is recovered immediately in solution (Fig. S9), preventing size-dependent activity investigations of the same peptide.

The strong correlation observed for μm−sized aggregation further emphasizes a fundamental common principle for infectivity enhancement of peptides: The hydrophobic surface properties may

not only be a driving force for peptides to form fibrils[56], increase fibril-fibril association[57,58] which leads to μm−sized aggregates, but also a prerequisite to bind and capture viral particles. An interplay between hydrophobicity, electric charge, and anions in buffer is well-known to control the formation of hierarchical structures, such as amorphous aggregates or amyloid fibrils[59–61]. Hydrophobicity as a prerequisite for efficient gene delivery was recently shown for polyplexes and polymers for non−viral applications[62,63]. The interaction of amphiphilic peptides with cellular lipid membranes was reported to be facilitated by electrostatic and hydrophobic interactions of the peptides[64] and multivalent binding to the amyloid surface reduced binding affinity to cells and infectivity of semen-derived enhancer of viral infection (SEVI) amyloid fibrils[65]. Thus, we conclude that, a high hydrophobicity can facilitate interaction of peptide fibrils with cell membranes and thereby increase infectivity enhancement as shown systematically in our peptide library.

Although, an alternating amphiphilic sequence pattern with hydrophobic and cationic amino acids is well-known for amyloids[66], and it is found in our library prevalent for active sequences, activity is not limited to these sequence motifs as long as the identified physicochemical properties are fulfilled. For example, we found the same relationship between microscopic aggregation and infectivity enhancement for structurally different endogenous short self-assembling amyloid peptides as exemplified by the three newly discovered highly effective peptides ISKLEYSNFSVRY, LANWMCLAKW, and VHDCVNITIK. Interestingly, these peptides are derived from different biological contexts. While the sequences ISKLEYSNFSVRY and LANWMCLAKW occur naturally in acyl phosphatase and human lysozyme, the peptide VHDCVNITIK is a sequence from the pathogenic human prion protein[52]. It is important to note that while these protein fragments can enhance infectivity, their physicochemical properties cannot be simply extrapolated to the parent protein, as shown exemplary for lysozyme and LANWMCLAKW (Fig. S40). However, it is not recommended to use peptides derived from the original pathogenic origin for therapeutic purposes because they may act as potential seeds for disease-related aggregation events. Amyloid fibrils that aggregate expose fewer terminal ends that are known to disrupt cellular membranes[67]. The fibrillar aggregates reported in this study do not show toxicity but good cell viability for HeLa cells (Fig. S2) as well as other cell types for similar fibrils, as reported elsewhere[32,68,69]. Microscopic aggregation has been previously observed in the non-diseased state of endogenous semen protein prostatic acid phosphatase[70], which suggests a broader biological significance of micrometer-sized aggregation for amyloids beyond simple disease-related associations.

Pioneering works have put much efforts in identifying amyloid fiber evolution mechanisms on a molecular and nm-size level, e.g. for oligo and protofilament formation[71–73]. On the inter-fiber mesoscale lateral aggregation was traced back H bonds[74] and VdW polar interaction modes[75] and to hydrophobic interactions[76,77]. Interestingly, experiments and simulations showed that dynamic, flexible fibrils are less prone to align into multi-fiber strands[37,78]. However, the exact structural determinants for the μm−sized aggregation of amyloid fibrils still remain elusive, and the majority of studies, reporting on mechanistic insights of amyloid formation, focus on aggregation in the nm−length scales (fibril formation)[79] and rarely on μm−sized aggregation. One of the reasons for the lack of studies that deal with understanding the multi-scale aggregation phenomena stems from the difficulty of relating molecular information to microscopic properties, which is especially challenging for self-assembly processes occurring within short time frames. Our data indicates that high β−sheet content of amyloidal peptides, and hydrophobicity are not only necessary for interaction with viral particles and cells but also favor aggregation of fibrils into μm−sized fibrillar aggregates. The higher surface hydrophobicity of CKIKIQI compared to CKIKQII which is observed from SFG measurements (Fig. 4B−v) may be

arising from a better intermolecular β−sheet packing of peptide monomers in the fibrils due to localization of the more hydrophobic amino acid isoleucine in the center of the peptide. Furthermore, a high propensity to form β−sheet structures does not necessarily lead to lateral aggregation of fibrils into microscopic aggregates[52]. For example, the sequence GNNQQNY forms β−sheet rich amyloid fibrils but with well hydrated external faces, which presumably hinder them to form microscopic aggregates (Table S7)[75]. The interplay between polar and non-polar interactions, so-called hydrophobic patches, at the fibril surface can determine the fate of ordered or less ordered fiber−fiber aggregation across length scales[36,80] and might explain why some amyloid fibrils form unordered large ($>10\,\mu m^2$) aggregates while others stay isolated or form well-aligned bundles. Our data shows that regular arrangement of polar and non-polar amino acids facilitates stable β−sheet rich structures and SFG measurements indicate that hydrophobic surface interface dominates for microscopically aggregating peptides. Therefore, surface hydrophobicity likely is a key driving force for formation and μm-sized aggregation[81] of amyloidal fibrils and an important characteristic besides charge−driven interactions to associate with viruses and cells[82]. The fewer water-accessible areas may result in higher fibril aggregation which may be at the same time the driving force for cell and virus interaction. However, structural evidence of these interactions is needed to support these hypotheses, which is beyond the scope of this study. Due to the lack of experimental methods to show the assembly dynamics and peptide−water interaction on very short timescales (∼ ns) which is necessary to explain molecular dynamics, our current efforts focus on molecular simulations which may connect structural information to physicochemical properties underlying intermolecular interactions which eventually result into microscopic aggregation.

The specific factors that determine the ordered or unordered aggregation of micrometer-sized particles are currently unknown, but they are important not only for understanding biological activity but also for a wide range of diseases. We believe that multi-scale and multiparameter considerations in mechanistic origins of microscopical aggregation are not only important for studying pathogenesis, such as neurodegenerative diseases but also for infectivity enhancement and understanding possible connections between them.

This study represents the first investigation of the multi-scale structure−property−activity correlation of self-assembling amyloidal peptides for in vitro retroviral transduction enhancement. In contrast to previous reports, we find that not the amyloid fibrils itself but rather their μm−sized aggregated states facilitate retroviral gene delivery. By applying a data-mining approach, we could perform a multi-parameter regression and unveil crucial co-dependencies in the peptide properties that govern their activity and act on length scales from nano to micrometers: Amphiphilic, hydrophobic sequences form microscopic aggregates with a high β−sheet content and a positively charged surface, all of that required for efficient retroviral transduction enhancement. The fibril-fibril aggregation and the fibril-virus interaction are both driven by the hydrophobicity and amphiphilicity of the peptide sequence. Importantly, we exploited these structure−property−activity findings to rationally create novel active sequences. Moreover, by successfully applying our findings to screen for active amyloid peptides from endogenous proteins, we validated the universal applicability of our study. In that way, our study contributes to a fundamental understanding on the emergence of bioactivity of microscopically aggregating amyloid fibrils.

## Methods
### Materials
Fmoc-protected amino acids and Wang resin were purchased from Novabiochem®. OymaPure®, *N*-ethyldi*iso*propylamine, piperidine and trifluoroacetic acid were obtained from Carl Roth. Dimethylformamide (DMF for peptide synthesis), diethyl ether and dimethylsulfoxid was

purchased from Acros Organics. Acetonitrile was purchased from Fisher Scientific. Uranyl acetate and fluorescamine were purchased from Merck. Lysozyme was purchased from Amresco. PBS, Thioflavin T and α-cyano-4-hydroxycinnamic acid were purchased from Sigma Aldrich. Proteostat® was purchased from Enzo Life Sciences. All chemicals are listed in Supplementary Data 1 and were used as received unless explicitly stated otherwise.

## Data-mining

The herein applied data-mining procedure covers data generation, processing, data extraction and evaluation, model development, prediction, and interpretation to evaluate the identified property–activity relationship.

The first step involved data generation, where 163 distinct peptides based on an infectivity-enhancing peptide EF-C (QCKIK-QIINMWQ) were synthesized (Table S2), six physicochemical and one biological properties were measured (see SI Section 1.1 and method section below). The collected data was then processed, as summarized in Table 1. Twelve bioinformatic properties including peptide net charge, hydrophobicity, isoelectric point, aliphatic index, hydrophobic moment index, boman index, instability index, graph shape index, upsilon steric parameter, smoothed upsilon steric parameter, polarizability, and normalized van der Waals volume were calculated via the "peptides" package in R (see SI Section 1.2)[40].

The next step was extraction and evaluation of the collected data: For an initial screening every descriptor was summarized in a table with the software Excel (Version 2304 Build 16.0.16327.20200, Microsoft 365) and imported to the data-mining software Orange3 (Orange3-v.3.34.0, Anaconda) to find the best correlating descriptor combinations via the functions "scatter plot → find informative projections" and "pearson correlations". The Pearson correlation coefficients for every descriptor were calculated via the function "correlation plot" of the software OriginPro 2021 v.9.8.0.200 (OriginLab). The property activity relationship was identified using the function "constraint multiple regression" analysis (constrained linear least-squares problems solution with the equation: $y = A0 + A1*x1 + A2*x2 + [...] + A5 * x5$) of the software OriginPro 2021 v.9.8.0.200 (OriginLab). The model was evaluated using various statistical metrics such as the Pearson correlation coefficient and the statistically significant descriptors were identified by $p < |t|$, $t = 0.05$.

The established linear regression model was validated with de novo peptides predicted from a pattern analysis technique and with existing peptides from a database screening.

The pattern analysis aims to find amino acid patterns and amino acids that are prevalent in high-active peptides (see SI Section 7). Briefly, amino acids were first classified into four categories based on their hydrophilicity and charge at pH 7.4 using the Kyte−Doolittle hydropathy scale[83]. The infectivity enhancement was categorized in low (< 4-fold virus only, <10% relative to EF-C), medium (10–70% relative to EF-C), high (> 70% relative to EF-C). All possible 7-mer patterns were then generated from these coarse-grained amino acids and matched against existing patterns in the peptide library. To identify patterns that are more prevalent in high-active sequences compared to medium- or low-active sequences, an activity index of a pattern $p$ ($AI(p)$, Eq. (1)) was calculated by subtracting the sum of its relative numbers in medium- and low-active sequences from its relative number in high-active sequences as in:

$$AI(p) = \frac{N_{\text{high}}(p)}{\sum N_{\text{high}}} - \frac{N_{\text{medium}}(p)}{\sum N_{\text{medium}}} - \frac{N_{\text{low}}(p)}{\sum N_{\text{low}}} \qquad (1)$$

For the amino acid analysis, the absolute abundance of amino acids in one category (low, medium, high) in the whole peptide library was first determined. The relative abundance of amino acids "Rel AA" was then calculated by determining the number "N" of peptides containing a certain amino acid "AA" relative to the total amount of peptides of one category (Eq. (2)).

$$Rel\ AA = \frac{N(peptides\ containing\ AA)_{\text{category}}}{\sum \text{peptide}_{\text{category}}} \qquad (2)$$

All pattern and amino acid analyses were conducted using Python scripts. Source code is provided[46].

The prediction and evaluation of new peptides were conducted by retranslating the coarse-grained best-performing patterns with best-performing amino acids and worst-performing amino acids to yield 10 peptide sequences designed to be active and 8 peptides designed to be inactive. To avoid human bias, the selection of amino acids from the pool of best-performing and worst-performing ones was conducted randomly.

For the data-base screening 538 short (< 20 amino acids) peptides across various sources (176 peptides from PDB[84], 260 peptides from a report[52] and 102 peptides from Waltz database[85]) were collected by their bioinformatic properties (hydrophobicity, isoelectric point, net charge, calculated the "peptides" package in R[40] v 4.0.3) and by any reported information on structure formation (TEM, FT-IR and ThT-fluorescence). A list of all considered peptides from the above-mentioned databases is summarized in Supplementary Data 2. Ultimately, 24 different peptides (SI Section 8) were selected based on fibril formation by TEM and preferably by net charge (> 0), hydrophobicity (> 0.2), β-sheet formation by ThT fluorescence.

## Solid-phase peptide synthesis and characterization

Peptide sequences were purchased from Phtd Peptides industrial Co. limited with purity of ≥ 95%. Some peptides were synthesized from us (marked with * in Table S3) by using an automated microwave peptide synthesizer (CEM, Liberty Blue™) at a 0.1 mmol scale using Fmoc protected Wang resin according to the standard coupling strategy. Briefly, the resin was swollen in DMF for 1 h and subsequently the Fmoc protecting group was cleaved with a piperidine solution (20 vol% in DMF) by microwaving at 155 W, 75 °C for 15 s and at 30 W, 90 °C for 50 s. The resin was washed three times with DMF and Fmoc protected amino-acid (5 equiv. relative to the resin loading capacity), DIC (5 equiv.) and Oxyma Pure® (10 equiv.) were dissolved in DMF and added to the reaction vessel. After microwaving at 170 W, 75 °C for 15 s and at 30 W, 90 °C for 110 s the resin was washed with DMF. Repeating this procedure for all required amino acids yields the desired peptide sequence. In the final step, the Fmoc protecting group was cleaved by microwaving at 155 W, 75 °C for 15 s and at 30 W, 90 °C for 50 s with piperidine solution (20% in DMF) and the resin was washed with DCM. The peptide was cleaved off the resin through treatment with 5 mL of trifluoroacetic acid/water/tri*iso*propylsilane (95% / 2.5% / 2.5%) for 2 h and precipitated in cold diethyl ether (40 mL). The precipitate was centrifuged at 3434 x g for 15 min and dissolved in water and 0.1% TFA and purified via HPLC using a gradient of water and acetonitrile containing 0.1% TFA or 0.1% NH₄OH for peptides containing mainly basic or acidic amino acids, respectively. HPLC runs for purification were conducted in preparative scale using a Shimadzu system with the following modules: DGU-20A5R, LC-20AP, CBM-20A, SPD-M20A, SIL-10AP, FRC-18A and a Phenomenex Gemini® 5 µm NX-C18, 110 Å, 150 ×30 mm, 5 µm pore size with a flow rate of 25 mL/min. The peptides were lyophilized to obtain a white solid. Purity (Fig. S35 and Table S3, Supporting Information) was analyzed by MALDI-TOF-MS (rapifleX MALDI-TOF/TOF, Bruker MALDI Synapt G2-SI, Waters) and HPLC-ESI-MS on a Shimadzu LC-2020 Single Quadrupole MS instrument equipped with the modules LC-20AD, SIL-20ACHT, SPD-20A, CTO-using a Kinetex EVOC18 100 Å LC 50 × 2.1 mm column with 2.6 µm pore size. An acetonitrile/water mixture with additional 0.1% formic acid was used as eluent.

**Table 1 | Summary of pre-processing methods to obtain output data for data-mining from the readout data of experimental descriptors**

| Descriptor | Method | Readout | Pre-processing | Output |
|---|---|---|---|---|
| Infectivity enhancement | β-galactosidase assay | β-galactosidase luminescence (infection rate intensity) in relative light units per second (RLU / s) | Calculation of n-fold infection in presence of peptides relative to virus only infection in absence of peptides. Calculation of ratio between n-fold infection of peptides relative to n-fold infection of EF-C (Infection Rel. EF-C). | Logarithmic infectivity relative to EF-C (Log$_{10}$ Rel Infect EF-C) |
| Fibril formation | TEM | Micrographs | Qualitative evaluation of high aspect ratio nanostructures (fibrils) | Fibril formation observed (1) or not observed (0) |
| Monomer to nanostructure conversion | Conversion Rate | Fluorescamine fluorescence intensity | Fluorescence intensity ratio before and after filtration | Percentual fluorescence intensity ratio (conversion rate) |
| Amyloid structure | ThT-fluorescence | Thioflavin-T fluorescence intensity | N-fold fluorescence intensity of peptides compared to buffer control sample | ThT-active (N-fold >2), or ThT-inactive (N-fold <2). |
| β-sheet content | FT-IR | Infrared absorbance (1300–1800 cm$^{-1}$) of peptide samples | Data-smoothing, normalization, calculation of second derivate of absorbance intensity, calculation of peak integral ratios between peaks at 1625-1635 cm$^{-1}$ (β-sheet) and approx. 1650-1700 cm$^{-1}$ (native structure). | Percentual β-sheet content |
| Charge | Zeta-Potential | Zeta-Potential in mV | - | Zeta-Potential in mV |
| Microscopic aggregation | Light Scattering Count Rate | Light Scattering Count Rate in kilo counts per second (kcps) | - | Light Scattering Count Rate in kilo counts per second (kcps) |
| Microscopic aggregation | Widefield Microscopy | Microscopy images | Count area size and number of aggregates (>10 μm²) via 3D objects counter (ImageJ) | Aggregate area size and number in a microscopy image. |

## Nanofiber–formation

To induce fibril–formation, peptides were dissolved in DMSO (10 mg/mL stock solution) and diluted in PBS (DBPS w/o calcium chloride and magnesium chloride, Sigma Aldrich) to a final incubation concentration of 650 μM or 1 mg/mL as indicated in the text. Other concentrations described in the text were achieved by further dilution of preformed PNFs. The solution was incubated at room temperature for 24 h in order to obtain fibrils. Literature-derived PNFs were prepared according to literature procedures (see SI Section 7).

## Amyloid fibril characterization

**ATR–FT-IR.** For ATR FT–IR spectroscopy measurements, 200 μL of the respective 1 mg/mL PNF solution was lyophilized and the resulting powder used for measurement. All spectra were recorded on a Bruker Tensor 27 spectrometer with a diamond crystal as ATR element (PIKE MiracleTM, spectral resolution 2 cm⁻¹). Every sample was measured with 64 scans. Data were analyzed with OriginLab software and calculation of β–sheet content was conducted according to a previous report[26]. **TEM.** For TEM measurements 5 μL of PNF fibrils (1 mg/mL) were placed on copper grids coated with carbon and formvar layer (300 mesh, Plano GmbH). After 10 min incubation time, the grids were staining with 4% uranyl acetate solution for 2.5 min and washed with water. Measurements were performed on a Jeol 1400 electron microscope with 120 kV acceleration voltage. **CR Assay.** For the determination of the amount of peptide monomer converting into nanostructures the amine reactive dye fluorescamine was used analogous to a previous reported protocol[26]. Briefly, 100 μL of the preformed PNF solution was centrifuged in a Vivaspin 500 tube (3 kDa MWCO) to separate fibers from free peptide monomer (16200 x g, 4 °C, 45 min). The filtrate and 100 μL of the non-filtered PNF sample were lyophilized and dissolved in 30 μL DMSO. 10 μL of the filtrate and non-filtered DMSO samples were submitted in a black UV Star® 384 microliter well–plates (Greiner bio–one) and 3 μL of the fluorescamine solution (10 mg/mL, DMSO) was added to measure the fluorescence enhancement. After 20 min of incubation at RT, fluorescence was recorded on an Infinite® M1000 PRO microplate reader (Tecan) at $\lambda_{em}$ = 470 nm upon excitation at $\lambda_{ex}$ = 365 nm with 10 nm bandwidths and multiple reads per well (3×3). Measurements were repeated in triplicates and averaged with standard deviation. All values were calculated as n-fold fluorescence enhancement (DMSO only as a reference was set to 1). The conversion rate CR was defined according to the equation Eq. (3).

$$CR = 100 - \left( \frac{100 \times fluorescence\ intensity\ (filtrate)}{fluorescence\ intensity\ (original)} \right) [\%] \qquad (3)$$

**ThT–assay.** 4 μL of 1 mg/mL peptide fibrils solution is added to a 20 μL, 50 μM ThT–solution in PBS. The mixture is incubated for 15 min at RT to allow binding of ThT dye with β–sheet rich structures. For reference PBS (4 μL) was added instead of peptide fibril solution. The samples were placed in black UV Star® 384 microliter well–plates (Greiner bio–one). Fluorescence spectra were recorded on an Infinite® M1000 PRO microplate reader (Tecan) at $\lambda_{em}$ = 488 nm upon excitation at $\lambda_{ex}$ = 440 nm with 10 nm bandwidths and multiple reads per well (3×3). Measurements were repeated in triplicates and averaged with standard deviation. A peptide fibril was considered as ThT-active if the fluorescence intensity was at least twice as strong compared to the control (PBS containing 10% DMSO). **Zeta-Potential.** For the determination of the surface charge of aggregated peptides zeta-potential measurements were conducted with a Zetasizer Nano ZS, Malvern Instruments. Unless stated otherwise the peptides were dissolved from DMSO (10 mg/mL) in PBS (pH 7.4) to concentration of 1 mg/mL and incubated for 1d at RT. Just before the measurement the 60 μL of the peptides were further diluted in 600 μL KCl (aq, 1 mM) in a 1 mL disposable

folded capillary cells (DTS–1060, Zetasizer Nano series, Malvern). The zeta potential of the peptides was derived from the electrophoretic mobility based on the Smoluchowski formula. All measurements were averaged over two individually performed experiments and conducted in triplicates for each sample. **Aggregate analysis**. The derived count rate of scattered light at 633 nm, 173° from the zeta-potential measurement was used as information on the light scattering intensity and turbidity of the sample as an indicator for microscopic aggregation[35]. Furthermore, fluorescence microscopy was conducted by staining 1 μL preformed fibrils (1 mg/mL) with 9 μL ThT (50 μM). Brightfield microscopy was performed simultaneously to evaluate aggregation also for non ThT-active fibrils. The measurements were performed on a Leica DMi8 microscope with 10x air objective and equipped with a Leica MC170 HD camera with a filter setting $\lambda_{em} = 527/30$ nm upon excitation at $\lambda_{ex} = 480/40$ nm) and processed with the Leica LASX software. To quantify size and number of microscopic aggregation of particles > 10 μm² at least 3 independently measured images were analyzed via the 3D object counter analysis tool of ImageJ, v 1.53f51. **Sum–frequency generation**. The SFG experiments were carried out with a femtosecond Ti: Sapphire amplified laser system (Spitfire Ace, Spectra–Physics, 800 nm, 40 fs, 1 kHz) with 5 W output power. We used 2 W to pump an optical parametric amplifier (TOPAS, light conversion) with a non–collinear difference frequency generation stage to generate broadband IR pulses. Another 1 W of the laser output passed through an etalon to generate narrowband visible pulses (10 cm⁻¹). The incident angles of the visible and IR beams were 36° and 41° with respect to the surface normal, respectively. The visible (13 μJ) and IR (5 μJ) pulses were overlapped spatially and temporally at the sample position. Subsequently, the generated SFG pulse was guided into a spectrograph (Acton SP 300i, Princeton Instruments), and detected with an EMCCD (Newton, Andor Technology). To avoid IR absorption of water vapor, we purged $N_2$ for the measurement. All spectra were collected in the ssp (denoting s-, s- and p-polarized SFG, visible and IR, respectively) polarization combination. The spectra were normalized to the non–resonant signal taken from z–cut quartz after subtracting a background spectrum.

**Virus–Peptide Interaction**. TZM-bl cell line and R5-tropic HIV-1 stock plasmid (pBRNL4.39-92TH014) were obtained from the National Institutes of Health AIDS Research and Reference Reagent Program. Human wild-type HeLa cell line were obtained from abcam, ab260075, LOT: GR3292155-1. The cell lines were not further authenticated, and no commonly misidentified cell lines are used within this study. Infectious R5–tropic HIV-1 stocks and infectivity assays were prepared analogous to a previous report[26]. Briefly, 10,000 TZM-bl cells were seeded into 96-well plates in 180 μL DMEM one day prior to infection experiment. Peptides were incubated two days prior to experiment at 650 μM concentration in PBS to generate fibrils. One day before the experiment the peptide samples were diluted to 130 μM in PBS. On the day of the infection experiment, 40 μl of 130 μM, 26 μM, 5.2 μM and 0 μM peptide samples were mixed 1:1 (v/v) with 40 μL of HIV-1 (prediluted 1:125 (v/v) in cell culture medium), resulting in fibril concentration of 65 μM, 13 μM, 2.6 and 0 μM during virion treatment. After 10 min incubation to allow binding of virus particles to peptide fibrils, 20 μL of the virus-peptide fibril mixture were added to 180 μL of cells resulting in final cell culture peptide fibril concentrations of 6.5 μM, 1.3 μM, 0.26 and 0 μM, respectively. The effect of peptide fibrils (final concentration on cells 6.5, 1.3, 0.26, 0 μM) on HIV-1 infection was studied 3 days post infection via a luminescence assay for β–galactosidase, which was reported by TZM–bl cells upon infection. For the property–activity correlation a peptide concentration of 1.3 μM was used as a standard condition and EF-C was always used as a reference to compare infectivity enhancement between peptides. The HIV-1 infection assay was conducted in three technical replicates and reproduced at least once.

The cell viability after addition of peptides to TZM–bl cells was studied via the CellTiter–Glo assay. To this end, 10,000 cells were seeded and on the next day serial diluted peptides were added. After 3 days the supernatant was removed and 100 μL CellTiter–Glo Reagent 1:1 diluted in PBS was added. After 10 min 50 μL was transferred to white microplate and luminescence was recorded by Orion microplate luminometer.

Confocal laser scanning microscopy studies were performed for the visualization of the cell–peptide interaction. HeLa cells were seeded one day prior to conducting the assay (40,000/well) in an 8–well IBIDI slide. 4 μL of the preformed peptide fibrils (1 mg/mL) were diluted with 4 μL Proteostat (Enzo Life Science, 1 μL stock in 999 μL PBS) and further diluted with medium to receive a final peptide concentration of 20 μg/mL. The nucleus of the HeLa cells was stained with Hoechst 33342 (NucBlue™, Thermo Fisher Scientific). The peptide solution mixture was transferred to the HeLa cells and incubated for 30 min at 37 °C before washing three times with PBS. The interaction of fibril clusters with cells was monitored after 30 min incubation time on a Stellaris 8 confocal laser scanning microscope (Leica) equipped with a 20x air objective and laser excitation wavelength of 405 nm (Hoechst) and 561 nm (Proteostat).

### Statistics and reproducibility

All virus assays were conducted in technical triplicates. All tested peptides for the virus infectivity assay were conducted at four different concentrations (6.5 μM, 1.3 μM, 0.26 μM, 0 μM). To verify reproducibility of the experimental findings the infection assays were at least repeated independently in biological duplicates. The physicochemical characterization methods for peptide self-assembly and morphology, that are determined via CR Assay, ThT–Assay and Zeta-Potential have been conducted in technical triplicates with two independent repetitions for Zeta-Potential and ThT-Assay measurements. Unless stated otherwise the aggregate analysis via widefield microscopy and TEM analysis have been conducted once with at least three microscopy images recorded for each peptide sample. ATR–FT-IR measurements have been conducted once for each peptide sample. The sum–frequency generation experiments have been conducted in two independent repetitions. To ensure comparability within individual measurements, the reference peptide "EF-C" was included in all assays. All replication attempts were successful.

No statistical method was used to predetermine sample size. No data were excluded from the analyses and the experiments were not randomized because it is not relevant for this study, as infectivity enhancement is independent from the order of samples tested. The investigators were not blinded to allocation during experiments and outcome assessment because it is not relevant for this study, as the results from infection assays are read out automated via a multi-plate reader.

### Reporting summary

Further information on research design is available in the Nature Portfolio Reporting Summary linked to this article.

## Data availability

The authors declare that the main data supporting the findings of this study are available within the paper and its associated Supplementary Information. Source data for manuscript and supplementary figures are provided with this paper. Other databases that are used for this study are "The Protein Data Bank" (PDB, https://www.rcsb.org/)[84], a report by Fernandez-Escamilla, A.-M., Rousseau, F., Schymkowitz, J. and Serrano, L (https://doi.org/10.1038/nbt1012)[52] and the WALTZ-DB 2.0 database (http://waltzdb.switchlab.org/home)[85]. A summary of peptides considered for evaluation from these databases is provided in a supplementary file. Source data are provided with this paper.

## Code availability

The computer codes and the corresponding input data for amino acid and peptide pattern analyses are openly available at the following data repository https://doi.org/10.5281/zenodo.8079726[46].

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

## Acknowledgements

The study was funded by the Deutsche Forschungsgemeinschaft (DFG, German Research Foundation)—Projektnummer 316249678—SFB 1279

(A02, A03, A05, C01). A.D. acknowledges support by BiGmax, the Max Planck Society's Research Network on Big-Data-Driven Materials-Science. T.B. acknowledges support from the Emmy Noether program of the Deutsche Forschungsgemeinschaft (DFG).

## Author contributions

KK conceived the idea, designed, and carried out the experiments, analyzed data, performed the formal analysis, and wrote the original draft. LRW designed, and carried the experiments related to virus transduction, analyzed data, performed the formal analysis, and revised the manuscript. AD, XY, YN analyzed data, performed the formal analysis, and revised the manuscript. TB, CVS, JM, and TW conceptualized and supervised, provided the funding resources, and revised the manuscript. The final manuscript was approved by all authors.

## Funding

## Competing interests

J.M. and L.R.W. hold patents and K.K., L.R.W., C.V.S., J.M. and T.W. submitted a patent for application of peptide fibrils for retroviral transduction enhancement. The remaining authors declare no competing interests. The authors have no relevant non-financial interests to disclose.
