## [Peer Review File · Nature Communications]

Data-Mining Unveils Structure–Property–Activity Correlation of Viral Infectivity Enhancing Self-Assembling PeptidesReviewer #1 (Remarks to the Author):

The manuscript titled "Data-Mining Unveils Structure–Property–Activity Correlation of Viral Infectivity Enhancing Self-Assembling Peptides" by Kaygisiz et al., is interesting and informative. Here the authors have used a very extensive and systematic approach to identify the determinants that dictate the viral transduction efficiency of short peptides that form beta sheet rich macromolecular structures. Though a bunch of earlier studies (from the same lab and from other researchers in the field) with similar concepts exist, what makes this study different is the following:

1. An extended peptide library consisting of 163 peptides made by rationalized modification of the original peptide EF-C
2. Many previous studies compared structure-morphology- activity whereas here it is structure-property of the peptides/final structures-activity.
3. The authors used data mining approach to correlate multiple experimentally measurable parameters (TEM, Zeta potential, scattering, transduction efficiency) and bioinformatic analysis of the peptide library.

Overall, this study provides a general scheme to identify if a given peptide sequence will aggregate to form beta rich aggregate structure and if so, can it be used for viral transduction application.

Key results: The main take-home message from this study is that microscopically large beta sheet rich aggregates with positive surface charge is a potential material for viral transduction application.

However, the importance of positively charged beta sheet rich aggregates for mediating viral infectivity were also studied previously by Dai et al. 2015 PNAS, Sieste et al. 2021 Adv Funct materials, Kirti et al. 2019 ACS Biomaterials etc. In the current manuscript authors also correlate number of microscopic aggregates formed by the designed peptides to their efficiency in infecting the cells with virus.

Validity and Significance: Over all the data interpretation and conclusions are reasonable. In the Section 6 of the manuscript, the authors compared peptides from functional and pathogenic amyloids and tested their efficiency in enhancing viral transduction.

A point to note here is that in Nature these peptides are usually found as their extended sequence which might have a completely different biophysical property than a truncated peptide version. Hence it will be difficult to say if a peptide derived from functional or pathogenic amyloid which shows enhanced viral transduction, will also show the same effect in their native/extended form. Rather with this study what is clear is that short peptides despite the source of origin, their physicochemical property determines the activity, which in this manuscript is viral transduction.

Data and methodology: The data and methodology in the manuscript are appropriate for the current study.

Analytical approach: All the analytical approaches are valid for the current manuscript. However, the infectivity data lacks statistical analysis.

Experiments with SFG spectroscopy are outside the scope of my expertise.

Suggested improvements.

1. The information about the time scale of aggregate formation is unclear from the manuscript.
2. In the Experimental section, it is unclear if the virus is added to the peptide solution or to the aggregates.
3. It would be useful to know any information on the interaction of virus and the aggregated structures.

4. Many of the measurement of the peptide aggregation results are seems to be in the absence of virus, it would be interesting to know if the addition of virus changes these parameters. For example, the fibril/ aggregate morphology or size?
5. Please also comment in the manuscript how surface area of the formed aggregates affects the viral transduction? If you mechanically downsize the aggregates that are larger than 10 μm^2 can it increase the infectivity?
6. Do the authors assume all the different peptide aggregates tested in this manuscript interact with cell membrane in a similar manner? and can it have a say in the viral infectivity?
7. How long are the aggregate laden virus exposed to the cells for transduction?
8. Could the aggregates with or without the virus be internalized by the cells ?
9. "TOC: We report that viral infectivity is enhanced by hydrophobic β - sheet rich peptides that form positively charged μm -sized aggregates by applying multi-parameter and multi-scale property– activity correlation". This sentence is confusing. Please rephrase the sentence. Also, from the results I understand that the aggregates are beta sheet rich not the peptides.

Clarity and context: The manuscript is clear and informative.

References: Please check the reference for CR assay in the Experimental section.

The queries given above are some of the concerning issues of the manuscript. These need to be addressed carefully for improving the manuscript before publication.

Reviewer #2 (Remarks to the Author):

This submission contributed by Kaygisiz et al. applied data mining method into structure–property–activity relationship of transduction enhancing peptides. The authors found that μm -sized -sheet rich aggregates enhance infectivity. This manuscript was very well written and structured. Firstly, I am very impressed by the detailed interpretation, rigorous logic, and massive work. Secondly, the investigations on amyloid-like peptide nanofibers provided abundant explorations and elucidations for their structure-property relationships. Thirdly, the boosting scientific development can support data mining for amyloid-like peptide nanofibers. Fourthly, the authors found unique phenomenon relating amyloid fibrils. Therefore, I highly recommend the publication of this work in Nature communication after minor revision.

Here are some suggestions.

1. It is suggested that highlighting the data mining procedure in TOC and in article figures to distinguish this work with the ordinary research article. The current version seems missing the graphical description of this key procedure.
2. All the experimental results (such as each TEM image) are suggested to provided the reference source one by one. This kind of error presents more in SI document.
3. In Figure 4A, please provide all the eight kinds of possible combinations of physicochemical features. This is the root for the subsequent data mining section. Then, Figure 4 can be separated into two figures, not so large as the current version.
4. Actually, the zeta potential values for amyloid-like peptide nanofibers vary in a relatively large range. The standard deviations for zeta potential values are not shown in the current dataset. Especially in SI document, some of the standard deviations of infections are shown but some are not. This will leave doubts to the related research fellows.
5. Most important, the data mining is highlighted at the beginning of the title. However, the data mining procedure and evaluation are not evidently provided in the whole article. Please display the data mining procedure in both abstract, methods, results, discussion, and conclusion.
6. Please check all the abbreviations throughout the manuscript. For example, "CAR T-cell therapy", "HIV", "HIV-1", "TZM-bl cells", etc. The full name should be presented at the first time. The SI document is suggested to check through.

Reviewer #3 (Remarks to the Author):

The present manuscript provides a comprehensive analysis of a range of small peptides on their ability to enhance viral infectivity. This property could be of use for viral uptake related applications. Correlations are obtained for various parameters tested and it is concluded that a combination of factors must be met for efficient enhancement of viral infectivity. Some of these factors were known already, such as the need for positive charges. One main new discovery is that the size of the aggregates matters, with larger aggregates being more potent. This information might well prove useful for understanding the effect of amyloids as adjuvants and should be of value to the field of viral enhancement by aggregates. The novelty to the amyloid field as a whole is rather limited. Much emphasis has been placed on correlations of activity with hydrophobicity and other physical parameters. While solid insights are obtained, they are not necessarily novel. That is, extensive work by Dobson and many others has established nearly the same concepts for amyloid protein misfolding in a series of studies over 2 decades ago. This work does not appear to be cited and it diminishes the broader significance of the present study. There are also numerous proven algorithms available that have good predictive qualities for aggregation and amyloid formation, making the present study less novel.

One of the parameters that has also been considered in the literature, but which is not discussed here, is beta-sheet propensity, which leads to a preference for beta-branched side chains in many amyloids. This is only a minor concern, but it was somewhat surprising that it was not included. The paper categorizes aggregates into fibril forming and amorphous aggregate forming peptides. It should be noted that this cannot be correlated merely with intrinsic properties of peptide sequence alone and it is also difficult to decipher simply based on one experiment under one condition. The problem is that nearly all amyloids can form fibrils or more amorphous material, it just matters on the conditions. Sometimes they can co-exist on the same EM grid and sometimes only subtle changes in conditions can switch from one to the other. Sometimes the behavior can even be complicated by the exact purification protocol prior to dissolution. The detection can also be difficult in some cases. The image shown for amorphous aggregates could indeed be fully devoid of fibrils, but it could also contain lots of fibrils. Sometimes fibrils can appear after stripping less ordered proteins or peptide from their surface. Surface binding of non-fibrillar material can make ThT fluorescence highly problematic. The authors need to acknowledge this potential complication more clearly.

The analysis of small peptide fragments from naturally occurring pathogenic amyloids is useful in the sense that it shows how their model has predictive power. Unfortunately, the authors go beyond this point, which leads to several concerns. First, the study uses small peptide fragments only, which makes it harder to learn much about the actual pathological proteins or peptides. Second, there is little evidence that these amyloids act by being adjuvants *in vivo*. In fact, it seems that nearly all peptides are not very potent at promoting infection based on the data obtained. Third, it is argued that the approach of looking for micron-sized aggregates of amyloid fibrils is often ignored in the field of pathogenic amyloids and that studying intrinsic properties of peptides (as done in the present study) could be helpful. I am not so convinced by this statement, as many studies have extensively looked at the correlation between amyloid size and toxicity. In the vast majority of cases, it has been concluded that the large, micron-sized aggregates are rather inert (so the exact opposite of the present study) and that smaller oligomers or protofibrils are more toxic. In this context, it is also important to note that the same protein can often take on many different aggregation states from smaller oligomers, short fibrils, all the way to micron-sized bundled fibrils of different polymorphs. There clearly maturation factors at play here that allow the same sequence to take up many different conformations and aggregation state and many of those will depend on cellular context. Another important practical consideration regarding the use of pathogenic amyloids as adjuvants would be the potential danger that these peptides might act as seeds and cause neurodegenerative or other diseases.

The discussion about the membrane interaction appears highly speculative. There is no structural information obtained for any of the peptides, and it is not clear which regions interact with the membrane. Much hydrophobic surface might be covered by inter filament interactions and is perhaps not even available for membrane interaction. Yes, the Wimley-White scales indicate that hydrophobic side chain have an easier time at partitioning into the membrane and such residues can clearly promote membrane binding. Having said that some peptides, such as TAT, can promote cell entry through the use of positively charged clusters. I suggest making this section

more convincing or take it out. It appears far too speculative as written.

Point-by-point response from the authors

We thank the reviewers for carefully evaluating our manuscript and providing constructive feedback. In the following, we will address the questions raised by the reviewers in a point-by-point fashion. Any changes made to the manuscript are highlighted in blue in the revised version and quoted in this response letter for reference. The reviewers' comments have helped to significantly improve the manuscript and we hope that our revised version is now suitable for publication in Nature Communications.

Black with grey background: Comments from the reviewer.

Black: Response of the authors.

Blue: Revisions to the manuscript.

Reviewer #1 (Remarks to the Author):

The manuscript titled "Data-Mining Unveils Structure–Property–Activity Correlation of Viral Infectivity Enhancing Self-Assembling Peptides" by Kaygisiz et al., is interesting and informative. Here the authors have used a very extensive and systematic approach to identify the determinants that dictates the viral transduction efficiency of short peptides that form beta sheet rich macromolecular structures. Though a bunch of earlier studies (from the same lab and from other researchers in the field) with similar concept exists what makes this study different is the following:

1. An extended peptide library consisting of 163 peptides made by rationalized modification of the original peptide EF-C
2. Many previous studies compared structure-morphology- activity whereas here it is structure-property of the peptides/final structures-activity.
3. The authors used data mining approach to correlate multiple experimentally measurable parameters (TEM, Zeta potential, scattering, transduction efficiency) and bioinformatic analysis of the peptide library.

Overall, this study provides a general scheme to identify if a given peptide sequence will aggregate to form beta rich aggregate structure and if so, can it be used for viral transduction application.

Key results: The main take-home message from this study is that microscopically large beta sheet rich aggregates with positive surface charge is a potential material for viral transduction application. However, the importance of positively charged beta sheet rich aggregates for mediating viral infectivity were also studied previously by Dai et al. 2015 PNAS, Sieste et al. 2021 Adv Funct materials, Kirti et al. 2019 ACS Biomaterials etc. In the current manuscript authors also correlate number of microscopic aggregates formed by the designed peptides to their efficiency in infecting the cells with virus.

Validity and Significance: Over all the data interpretation and conclusions are reasonable. In the Section 6 of the manuscript, the authors compared peptides from functional and pathogenic amyloids and tested their efficiency in enhancing viral transduction. A point to note here is that in Nature these peptides are usually found as their extended sequence which might have a completely different biophysical property than a truncated peptide version. Hence it will be difficult to say if a peptide derived from functional or pathogenic amyloid which shows enhanced viral transduction, will also show the same effect in their native/extended form. Rather with this study what is clear is that short peptides despite the source of origin, their physicochemical property determines the activity, which in this manuscript is viral transduction.

Data and methodology: The data and methodology in the manuscript are appropriate for the current study. Analytical approach: All the analytical approaches are valid for the current manuscript. However, the infectivity data lacks statistical analysis. Experiments with SFG spectroscopy are outside the scope of my expertise.

We thank the reviewer for carefully reading our manuscript and the positive evaluation of the novelty and relevance of our work. Concerning the reviewers' point about the use of model peptides derived from different proteins, we would like to refer the reviewer to our respective discussion in our answer to question 5 of reviewer #3.

Suggested improvements.

Question 1. The information about the time scale of aggregate formation is unclear from the manuscript.

Answer 1. We thank the reviewer for pointing this out. To clarify this, we have added **Figure S4** and the following information to the manuscript and SI.

Manuscript, **P7**: The timescale for the aggregation of peptides is within seconds as can be observed via TEM and turbidity from light scattering (**Figure S4**).

SI, **P8**: Fibril formation and μm -sized aggregation occur immediately upon incubation of peptides in PBS as demonstrated exemplary for CKFKFQF with TEM micrographs (**Figure S4A, B**) and by turbidity as observed via bare eye (data not shown), and light scattering exemplary shown for CKFKFQF (**Figure S4C**) and QCIKIQINMWQ (**Figure S4D**). No significant change in light scattering count rate is observed regardless of measurement directly after sample preparation (T_0) or after incubation for 1 day (T_{1d}) or 2 months (T_{2m}), indicating that the fibril formation and aggregation is completed within the timeframe of sample preparation (within seconds).

SI, P8: Figure S4

Figure S4 Time dependent fibril-formation and aggregation of selected peptides CKFKFQF and QCKIKIQINMWQ. **A** Schematic illustration and TEM micrograph of T₀ incubation condition. The peptide monomer, dissolved in DMSO (10 mg/mL), was added to a PBS droplet on a TEM grid and incubated at 1 mg/mL concentration for 5 min. Fibril formation is observed via TEM (scale bar 1 μm). **B** Schematic illustration and TEM micrograph of T_{1d} incubation condition. The peptide monomer, dissolved in DMSO (10 mg/mL), was added to PBS and incubated at 1 mg/mL concentration for 24 h. Then, the peptide fibrils were incubated on a TEM grid for 5 min. Fibril formation was observed via TEM (scale bar 1 μm). **C** The count rate of scattered light was determined from Zeta-Potential measurements for CKFKFQF (0.1 mg/mL) after T₀, T_{1d}, and T_{2m}. For T₀ incubation condition, the peptide was diluted from DMSO (10 mg/mL) in PBS (1 mg/mL) and immediately further diluted in 1 mM KCl to 0.1 mg/mL for measurement, which took approximately 3 min. For T_{1d} and T_{2m} measurements, the sample was incubated at 1 mg/mL concentration for one day and two months, respectively, and diluted in 1 mM KCl to 0.1 mg/mL just before measurement. **D** The count rate of scattered light was determined from Zeta-Potential measurements for QCKIKIQINMWQ (0.1 mg/mL) after T₀, and T_{2m}.

Question 2. In the Experimental section, it is unclear if the virus is added to the peptide solution or to the aggregates.

Answer 2. We thank the reviewer for drawing attention to this point and have added the following information in the experimental section.

Manuscript, **P24:** Briefly, 10,000 TZM-bl cells were seeded into 96-well plates in 180 μL DMEM one day prior to infection experiment. Peptides were incubated two days prior to experiment at 650 μM concentration in PBS to generate fibrils. One day before the experiment the peptide samples were diluted to 130 μM in PBS. On the day of the infection experiment, 40 μL of 130 μM, 26 μM, 5.2 μM and 0 μM peptide samples were mixed 1:1 (v/v) with 40 μL of HIV-1 (pre-diluted 1:125 (v/v) in cell culture medium), resulting in fibril concentration of 65 μM, 13 μM, 2.6 and 0 μM during virion treatment. After 10 min incubation to allow binding of virus particles to peptide fibrils, 20 μL of the virus-peptide fibril mixture were added to 180 μL of cells resulting in final cell culture peptide fibril concentrations of 6.5 μM, 1.3 μM, 0.26 and 0 μM, respectively.

Question 3. It would be useful to know any information on the interaction of virus and the aggregated structures.

Answer 3. In previous studies from us (**Figure I A, B**) and others (**Figure I C**), the interactions of selected peptide fibrils with viruses via electron and confocal microscopy was investigated and it was found that viruses are associated to the fibrillar aggregates.¹⁻³ To clarify these known interaction modes, we included the following passage in the introduction of the manuscript.

Manuscript, **P3**: In previous studies we and others have shown via electron and fluorescence microscopy that virus-particles associate to fibrillar structures.^{12,26,27}

Figure I A Confocal microscopy measurements of EF-C (QCKIKQIINMWQ, red) before and after virus (green) addition, scale bars 15 μ m. Microscopy measurement adapted from Yolamanova et al.³ **B** Fibrillar KFKFQFNMWQ peptide morphology before (TEM, 1 mg/mL in PBS, stained with Uranylacetat) and after (Cryo-EM, 20 μ g/mL) addition of virus like particle (black arrow), unpublished results, scale bars 100 nm. **C** TEM images of α -Syn fibrils before and after addition of virus (black arrows), scale bars 200 nm. EM measurements adapted from Kirti et al.⁴

Question 4. Many of the measurements of the peptide aggregation results are seems to be in the absence of virus, it would be interesting to know if the addition of virus changes these parameters. For example, the fibril/ aggregate morphology or size?

Answer 4. There are no visible changes upon the addition of virus particles as qualitatively observed and depicted in **Figure I A**³ via fluorescence microscopy and for fibril morphology observed from us (**Figure I B**, unpublished) and others (**Figure I C**)⁴ via electron microscopy.

We included this piece of information now in the manuscript.

Manuscript, **P18**: Noteworthy, there are no obvious changes in fibril formation and aggregation properties of the fibrils in the presence of virions, based on our own data and earlier reports.^{12,55}

Question 5a. Please also comment in the manuscript how surface area of the formed aggregates affects the viral transduction?

Answer 5a. We thank the reviewer for giving us the opportunity to further improve our aggregate size and number correlation analysis. We have now re-analyzed the aggregation in triplicate measurements and correlated the number, median size area, average size area and total surface area of all visible aggregates observed via widefield microscopy. Representative microscopy images are now shown in **Figure S32** and the quantitative analysis is summarized in **Table S8**. We have additionally plotted these new data in **Figure S8** to demonstrate how the surface area of the formed aggregates affects viral transduction.

We have observed that the light scattering count rate as well as the infectivity does not correlate with an increasing surface area but rather with the increasing number of visible aggregates (**Figure S8 I**). The presence of few (approx. < 100 aggregates in 1330 $\mu\text{m} \times 1330 \mu\text{m}$), large (> 100 μm^2) aggregates does not enhance infectivity, but a rather high number (approx. > 100 aggregates in 1330 $\mu\text{m} \times 1330 \mu\text{m}$) of smaller (> 10 μm^2) visible aggregates correlate with increased infectivity (**Figure S8 II, III**). We hypothesize that either too few virions can associate with the larger fibril aggregates or that many smaller aggregates can transduce many cells more efficiently than few larger aggregates. This is supported by the consideration of cumulative surface area of the particles, which eliminates the distribution but only considers total surface area (**Figure S8 IV**), that does not correlate with the count rate or infectivity.

We have included this in the Manuscript and SI.

Manuscript on **P18**: The importance of well-distributed aggregates rather than aggregate size is emphasized by findings from cumulative surface area of total aggregates, where we do not observe any difference between the surface area of “active” aggregates and “inactive” aggregates (**SI Section 4.3, Figure S8**).

SI, P11: The light scattering count rate as well as the infectivity enhancement correlate with increasing number of visible aggregates (**Figure S8 I**, $R = 0.84$ for Log_{10} Count Rate correlation with Log_{10} Aggregate Number (**Figure S8 AI**). In contrast, having few larger aggregates (> 100 μm^2) does not enhance infectivity as can be observed from the negative correlation between infection rates and count rates in **Figure S8 II, III**. We hypothesize that either too few virions can associate with the larger fibril aggregates or that many smaller aggregates can transduce many cells more efficiently than few larger aggregates. This is supported by the correlation of the cumulative surface area of the particles that, instead of the full distribution, only considers total surface area (**Figure S8 IV**), and shows no correlation with the count rate or infectivity.

SI, P11, Figure S8

Figure S8 Graphical summary of aggregate size and number analysis as correlation plots with count rate and infection rate. The aggregate analysis was performed in triplicate measurements (Std. Dev.) by automated 3D objects counting function (ImageJ) from widefield microscopy images (**Figure S32**), area 1330 μm × 1330 μm. The count rate and infection rate data are retrieved from **Table S8**.

Question 5b. If you mechanically downsize the aggregates that are larger than 10 μm² can it increase the infectivity?

Answer 5b. We thank the reviewer for this excellent suggestion to evaluate the impact of mechanical downsizing aggregates. We have exemplary tested the peptide CKFKFQF from our library. In order to study the impact of aggregate size on infectivity, one has to ensure that the same peptide sequence is applied and mechanically treated in a way, that it shows same chemical composition but different physical aggregation behavior. Applying mechanical pressure on the peptide fibrils is a reliable way to downsize aggregates (**Figure S9 All, BII, C**). Interestingly, once the physical pressure is removed, the initial aggregate size is recovered within seconds (**Figure S9 AllI, BIII,**

C). We therefore propose that it is not possible to achieve different aggregate sizes for the same peptide sequence by simple mechanical forces. This observation is further emphasized by ultrasonication experiments, which show comparable aggregate number and size before and after ultrasonication (**Figure S9D**) and similar infectivity values for peptides treated and not treated with ultrasonication (**Figure S9E**). We pointed this out in the manuscript and SI.

Manuscript, **P18** Interestingly, the formation of certain aggregate size and number appears to be an intrinsic feature of the non-covalent self-assembly properties of the molecular sequence, e.g. after mechanical downsizing aggregates by pressure or ultrasonication, their original size is recovered immediately in solution (**Figure S9**), preventing size-dependent activity investigations of the same peptide.

SI, **P12:** Mechanical downsizing of μm -sized aggregates can be achieved by applying pressure on a peptide solution which is entrapped between two glass slides (**Figure S9 AII, BII, C**). However, upon releasing the physical pressure, the initial aggregate size is recovered within seconds (**Figure S9 AIII, BIII, C**). This fast recovery likely originates from the fast non-covalent self-assembly of the peptide structures. This is emphasized also by ultrasonication experiments, which show comparable aggregation before and after ultrasonication (**Figure S9 D**) and similar infectivity values for peptides treated and not treated with ultrasonication (**Figure S9 E**). Thus, it is not possible to tune the aggregate sizes under comparable conditions for the same peptide sequence by simple mechanical forces.

SI, P12: Figure S9

Figure S9 Changes in aggregate size upon mechanical stress. **A** Schematic illustration of I aggregates composed of self-assembling peptide fibrils placed on a microscope objective and covered by a glass cover slip. **II** Applying pressure on glass cover slip mechanically downsizes aggregates, **III** however, once the pressure is released (e.g. by lifting the cover slip) the initial aggregate size is recovered. **B** Brightfield microscopy measurements (objective 20x) of CKFKFQ (1 mg/mL, 30 μL) covered by a glass cover slip, **I** before applying pressure, **II** after applying pressure and **III** after releasing pressure by lifting cover slip, scale bar 100 μm. **C** Aggregate size distribution for objects > 10 μm² in an area of 1331 μm × 1331 μm, error bars indicate std dev. from triplicate measurements. **D** Applying ultrasonication (US) on CKFKFQ (1 mg/mL in PBS) for 5 min or 1 h does not affect infection rates or **E** aggregate size and number compared to no ultrasonication, brightfield microscopy scale bar 100 μm.

Question 6. Do the authors assume all the different peptide aggregates tested in this manuscript interact with cell membrane in a similar manner? and can it have a say in the viral infectivity?

Answer 6. All peptides which enhance viral infectivity do this by facilitating the colocalization of virus and cells as evidenced by fluorescence microscopy measurements, studied extensively in previous reports for EF-C³ and also shown in this study for all tested peptides. The attachment of virions to the cell membrane is the major limiting step for infections and increasing the numbers of virions at the cell surface results in higher cell entry, which in turn enhances infection rates. While the main driving force for this interaction has been believed to be electrostatic attraction^{2,4-6} of positively charged fibrils binding to negatively charged virions and cellular membranes, here, using a multi-parameter correlation analysis, we found that it also considers contributions from hydrophobic interactions. The importance of hydrophobicity can be illustrated with the following examples: Previously we found that hydrophobic, negatively charged fibrils can bind to viruses, but not to cells.^{1,2} In the current study we presented some peptides (**Figure S15**) such as CEIEIQINMWQ that induce **weak** infectivity enhancement because they are **strongly** negative charged fibrils with **high** hydrophobicity. However, in another recent study currently submitted for publication,⁷ we found via machine-learning algorithm examples of peptides with **good** infectivity enhancement for **moderately** negative charged fibrils with **high** hydrophobicity.⁷ This shows that even fibrils without positive Zeta-Potential can enhance viral infectivity, if strong hydrophobic interactions are present. The infectivity and thus the efficiency of association of fibrils with virus and cell membrane is guided by the balance between electrostatic and hydrophobic interactions. Furthermore, all active peptides have to form aggregates, to enable also mechanical interaction points for entangling with protrusions of cells.⁸ This property is likely driven by hydrophobic interactions of fibrils resulting in aggregate formation and therefore a common feature underlying both cell-membrane interaction and aggregate formation.

Therefore, we believe that only considering fibril–cell-membrane interaction as a function of electrostatic interactions is an oversimplification and fibril-cell interactions are rather governed by an intricate balance between multiple intermolecular forces, such as electrostatic and hydrophobic contributions. Thus, the main interaction mode of each peptide aggregate depends on the balance of electrostatic and hydrophobic properties and is determined for each peptide separately depending on their physicochemical properties.

We included a short remark on our recent report supporting our findings.

Manuscript, **P11**: We hypothesize that these peptide fibrils can slightly enhance infectivity because of their extraordinarily high hydrophobicity (> 0.6) and microscopic aggregation of fibrils, which favor interactions with viruses and cells despite the electrostatic repulsion. The importance of considering hydrophobic interactions in addition to electrostatic attraction is further emphasized by a recent report from us, in which we applied machine learning approach to target different sequence spaces.⁴⁴

Question 7. How long are the aggregate laden virus exposed to the cells for transduction?

Answer 7. We thank the reviewer for pointing this missing information out. To clarify this, we added the following information in the experimental section.

Manuscript, **P24:** The effect of peptide fibrils (final concentration on cells 6.5, 1.3, 0.26, 0 μ M) on HIV-1 infection was studied **3 days post infection** via a luminescence assay for β -galactosidase, which was reported by TZM-bl cells upon infection.

Question 8. Could the aggregates with or without the virus be internalized by the cells ?

Answer 8. We previously investigated the cell-uptake of amyloid-type peptide additives and would like to kindly refer the reviewer to a previous publication from us: Yolamanova et al. showed in Figure 4³ and Figure S9³ the uptake of peptide aggregates in HeLa cells with viruses³ and Schütz et al. showed in Figure 4⁸ the uptake of peptide aggregates in HeLa cells without viruses.⁸ In both cases the peptide aggregates are internalized by cells in a comparable fashion regardless of the presence of viruses.

Question 9. "TOC: We report that viral infectivity is enhanced by hydrophobic β -sheet rich peptides that form positively charged μ m-sized aggregates by applying multi-parameter and multi-scale property– activity correlation". This sentence is confusing. Please rephrase the sentence. Also, from the results I understand that the aggregates are beta sheet rich not the peptides.

Answer 9. We thank the reviewer for giving us the opportunity to improve the TOC. We rephrased the sentence and graphically changed the illustration for a better understanding.

TOC, **P2:** **By applying a multi-parameter and multi-scale property–activity correlation in a data-mining approach, we find that hydrophobic β -sheet rich peptides that form positively charged μ m-sized aggregates are efficient enhancers of viral infectivity.**

For the changed TOC we would like to refer to our answer to question 1 of reviewer #2.

Clarity and context: The manuscript is clear and informative.

Question 10. References: Please check the reference for CR assay in the Experimental section.

Answer 10. We thank the reviewer for drawing attention to the wrong reference and corrected the reference now.

Manuscript, **P23:** [...] fluorecamine was used analogous to a previous reported protocol.²⁶

The queries given above are some of the concerning issues of the manuscript. These need to be addressed carefully for improving the manuscript before publication.

Reviewer #2 (Remarks to the Author):

This submission contributed by Kaygisiz et al. applied data mining method into structure–property–activity relationship of transduction enhancing peptides. The authors found that μm -sized β -sheet rich aggregates enhance infectivity. This manuscript was very well written and structured. Firstly, I am very impressed by the detailed interpretation, rigorous logic, and massive work. Secondly, the investigations on amyloid-like peptide nanofibers provided abundant explorations and elucidations for their structure-property relationships. Thirdly, the boosting scientific development can support data mining for amyloid-like peptide nanofibers. Fourthly, the authors found unique phenomenon relating amyloid fibrils. Therefore, I highly recommend the publication of this work in Nature communication after minor revision.

We thank the reviewer for the positive evaluation and strong endorsement of our manuscript.

Here are some suggestions.

Question 1. It is suggested that highlighting the data mining procedure in TOC and in article figures to distinguish this work with the ordinary research article. The current version seems missing the graphical description of this key procedure.

Answer 1. We thank the reviewer for pointing this out and giving us the possibility to improve the TOC. We have now changed the graphical illustration and the caption for a better understanding.

Manuscript, P2, TOC:

TOC By applying a multi-parameter and multi-scale property–activity correlation in a data-mining approach, we find that hydrophobic β -sheet rich peptides that form positively charged μm -sized aggregates are efficient enhancers of viral infectivity.

Question 2. All the experimental results (such as each TEM image) are suggested to provided the reference source one by one. This kind of error presents more in SI document.

Answer 2. We have now added the corresponding SI references for the summarized experimental results in the manuscript and SI.

The added references in the **manuscript** are on:

P5, Figure 1 Caption: Experimental methods, such as TEM (**Figure S33**), zeta-potential, ThT fluorescence (**Table S5**), turbidity analysis (**Table S8**) as well as bioinformatic characterization such as hydrophobicity, net charge and isoelectric point (pI) (**Table S4**) were applied to correlate with **D** the infectivity enhancement of HIV-1 on TZM-bl cells by the peptides, as quantified via a chemiluminescence based assay (**Table S5**). HIV-1 and peptides were incubated together before they were added to cells.

P7: [...] we additionally characterized regarding the conversion of monomers to fibrils (**Figure S5**, conversion rate, $R = 0.50$, **Figure S7 B**, **Table S5**).

P7: Plotting the zeta-potential against Log_{10} Infection relative to EF-C shows indeed the strongest linear correlation ($R = 0.67$, **Figure C**, **Figure S5**, **Table S5**).

P7: Another comparably high correlating property is microscopic aggregation of the peptides, which was measured via the count rate of scattered light during zeta-potential measurement (Log_{10} Count Rate, $R = 0.65$ **Figure D**, **Figure S5**, **Table S5**, **Table S8**).

P7: [...] by fluorescence microscopy (**Figure E**, **Figure S32**) correlates linearly with the logarithmic scattering intensity (**Figure 2F**, **Table S8**), which further validates [...].

P7: Despite the high correlation of zeta-potential and count rate with infectivity enhancement (**Figure C**, **D**, **Table S5**),[...].

P8, Figure2 Caption: **B** Histogram of Log_{10} Infection Rel EF-C showing ThT-active (red) and ThT-inactive (blue) peptide distribution (**Table S5**). **C** Zeta-potential plotted against Log_{10} Infection Rel EF-C. Error bars indicate standard deviation from triplicate measurements (**Table S5**). Linear fit with Pearson correlation factor $R = 0.67$. **D** Count rate of scattered light plotted against Log_{10} Infection Rel EF-C. Linear fit with Pearson correlation factor $R = 0.65$. Error bars indicate standard deviation from triplicate measurements (**Table S5**). **E** Selection of peptides with high and low count rate showing relationship between number of particles and count rate of scattered light (**Table S8**). Peptide fibrils were preincubated at 1 mg/mL one day and diluted to 0.1 mg/mL with ThT solution (50 μM) just before measurement (**Figure S32**). Number of aggregated fibrils ($>10 \mu\text{m}^2$) were quantified via fluorescence microscopy in an area of $1330 \mu\text{m} \times 1330 \mu\text{m}$ in triplicate measurements via automated counting (3D object counter ImageJ), scale bar 200 μm . **F** Log_{10} Count rate from zeta-potential measurements of representatively selected peptides plotted against averaged number of aggregated particles in an area of $1330 \mu\text{m} \times 1330 \mu\text{m}$ (**Table S8**). Error bars indicate standard deviation from triplicate measurements, linear fit (solid line) demonstrates a high correlation with a Pearson R of 0.84.

P9: [...] visualizes that a positive zeta-potential, high count rate and high hydrophobicity are coinciding with infectivity enhancement (**Figure 3A**, **Table S5**).

P9: [...] compared to the single features alone, an increase by 22% compared to best correlating single feature zeta-potential ($R = 0.67$) (**Figure 3B**, **Table S5**).

P9: However, in combination with all other calculated descriptors the correlation coefficient increases by roughly 70% (**Figure S11**, $R = 0.67$, **Table S4**).

P10, Caption Figure 3: **A** Four-dimensional plot visualizes that a positive zeta-potential (**Table S5**), high hydrophobicity (**Table S4**), and high count rate of scattered light (**Table S5**) are corresponding to infectivity enhancement (color coded Log_{10} Infection Rel. EF-C, **Table S5**). **B** Multiple linear regression (constrained linear least-squares problems solution) with the equation: $y = A_0 + A_1 * x_1 + A_2 * x_2 + \dots + A_5 * x_5$ with best performing single parameters of 163 peptides based on EF-C (**Figure S5**, **Table S4**, **S5**):

P10: The most prevalent case (38% of the library, 75% of all active peptides, Table S5) for infectivity enhancement are peptides assembling into fibrils with β -sheet secondary structure, [...]

P11: These peptides contain a high amount (> 20%) of hydrophobic amino acids such as W or Y and form amorphous amyloid aggregates with high β -sheet content (Table S5).

P11: Only weak infectivity enhancement is observed for peptides with negative zeta-potential even if the fibrils with high β -sheet content are aggregating (Figure 4A–3, Table S5). However, as it also can be extracted from the single parameter correlation with zeta-potential (Figure 2A, Table S5). [...]

P13, Figure 4 Caption: Evaluation of global property-activity relationship for infectivity enhancement with selected peptides as case studies to gain insight into the mode of action. Internal order is evaluated by TEM (Figure S33, Table S5, fibril formation) and FT-IR (Figure S34, Table S5 β -sheet content), surface charge by zeta-potential and infectivity is shown in absolute infection rates (Table S5). [...] **B** Comparison of physicochemical properties and activity of CKIKIQI and CKIKQII. **i** TEM micrographs and β -sheet content determined by FT-IR (Figure S34), scalebar 1 μ m. **ii** Fluorescence microscopy of ThT-stained peptide fibrils with their respective light-scattering count rate (Table S5), scalebar 200 μ m.

P15: The majority (55%) of the active sequences and 44 peptides in the library (163 peptides, Table S3) were [...].

P15: Eight of the 10 newly created peptides composed of the identified pattern and amino acids prevalent in highly active sequences show infectivity enhancement (Figure 6B, Table S6).

P15: Strikingly, in this way a cysteine- and glutamine-rich peptide KCQCQCQ was found, which showed activity in the range of EF-C while being half the length of EF-C (Figure 6B, Table S6).

P16, Caption Figure 6: [...] and side chain charge at pH 7.4 (+ = positive, - = negative, 0 = neutral) enables identification of most abundant sequence pattern in highly active peptides (Table S5) that is an alternating amphiphilic motif P⁺H⁰P⁺H⁰P⁰H⁰P⁰ and P⁺H⁰P⁰H⁰P⁰H⁰P⁰. **B** New peptides are created by combining the preserved pattern P⁺H⁰P⁺H⁰P⁰H⁰P⁰ with the identified best-performing amino acids. 8 of 10 newly created peptides are enhancing infectivity (> 10% Infection Rel EF-C at 1.3 μ M, Table S6) and overall confirm the identified property-activity relationship. **C** Graphical visualization of the most important property-activity relationship zeta-potential plotted versus Log₁₀ Infection relative to EF-C. The color code represents the Log₁₀ Count Rate from light scattering, which can be traced back to microscopic aggregation (Table S6). Error bars indicate standard deviation from triplicate measurements. **D** Selected representative example for fibril morphology (TEM, Figure S25, scale bar 1 μ m) and [...]

P17, Figure 7 Caption: [...] Numbers at the end of bars indicate n-fold infectivity enhancement compared to virus-only infectivity (0 μ M) (Table S7). **B** Zeta-potential plotted against Log₁₀ Infection Rel EF-C. Color indicates Log₁₀ Count Rate by scattered light. Three peptides (box) that show identified important physicochemical properties that have positive zeta-potential, show fibril formation, and aggregation are enhancing infectivity (Table S7). Error bars indicate standard deviation from triplicate measurements. **C–E** TEM micrograph (1 mg/mL, PBS, Figure S36, S37 scalebar 100 nm) [...].

P18: [...] showed infectivity enhancement of retroviral transduction in the range of 30–40% relative to EF-C (Figure 7A, Table S7). These newly found transduction-enhancing peptides have a positive zeta-potential (Figure 7B, Table S7, +9.3 mV, +14.7 mV, +9.3 mV, respectively), microscopically aggregate (Figure 7B, Figure S29–B, Table S7), assemble into fibrils (Figure S36, S37, Figure 7C–E) with a high β -sheet amount (Figure S30, Figure S39, Table S7) and are ThT-active (Figure S29–A).

P20: For example, the sequence GNNQQNY forms β -sheet rich amyloid fibrils but with well hydrated external faces, which presumably hinder them to form microscopic aggregates (Table S7).

The added references in the SI are on

P3 and non-active peptides will be considered rather than comparing exact ThT fluorescence intensities (Table S5, S6, S7).

P3 [...] bright field and fluorescence microscopy (Figure 2, Figure S31, Figure S32).

P4 [...] Since the fibrils formed by the short-self assembling sequences are highly polymorphic (TEM, Figure S33) and form heterogenous aggregate size (microscopy, Figure S32), it is not straightforward to apply DLS to determine polydispersity of aggregates. [...] methods such as TEM (for nanostructure verification, Figure S33) and microscopy (for macroscopic verification, Figure S32).

P4: [...] All infectivity data discussed in this study are referring to an *in vitro* peptide concentration of 1.3 μM on cells (Table S5).

P5 :[...] smoothed upsilon steric parameter, polarizability, normalized van der Waals volume (Table S4).

P9, Figure S5, Caption: All columns except of Conversion Rate (CR) contain data of 163 peptides. Conversion rate was determined for 80 representative subset of peptides (Table S5). Abbreviations: Absolute Infectivity at 1.3 μM peptide concentration (Abs. Inf.), and the logarithmic infectivity relative to EF-C (Log_{10} Inf.), the derived count rate of scattered light from zeta-potential measurements (Log_{10} Count Rate), zeta potential (Zeta-Pot.), fibril formation (Fibril) as evaluated qualitatively from TEM measurements (Figure S33), conversion of monomers to assembled structure (Conversion Rate, CR), percentual β -sheet content from fourier transform infrared spectroscopy (FT-IR, Figure S34) measurements (β -sheet), categorical ThT fluorescence (a peptide is ThT-active, if n-fold ThT fluorescence is >2 , Table S5), Net charge (NC), hydrophobicity according to Fauchère scale (H_{fauchere}) and Kyte-Doolittle scale ($H_{\text{KYTE-DOOLITTLE}}$), isoelectric point (pI), aliphatic index (AI), hydrophobic moment index (HMI), boman index (BI), instability index (II), graph shape index (GSI), upsilon steric parameter (USP), polarizability (P), and normalized van der Waals Volume (Norm vdW Vol.) Table S4.

P10, Figure S6, Caption: Graphical representation of calculated hydrophobicity (Table S4) plotted versus log_{10} Infection Rel. EF-C (Table S5). The color code represents fibril formation (red) and no fibril formation (blue) as observed via TEM (Figure S33, Table S5).

P10, Figure S7, Caption: Characterization of β -sheet and monomer to assembly conversion of the peptide library (Figure S34, Table S5).

P13, Figure S10, Caption: Multiparameter analysis of assembly (Figure S33), secondary structure (Figure S34), surface charge and microscopic aggregation (Table S5).

P14, Figure S11, Caption: graph shape index (GSI), upsilon steric parameter (USP), polarizability (P), normalized van der Waals Volume (Norm vdW Vol.), Table S4.

P15, Figure S12, Caption: Multiparameter correlation of experimental and bioinformatic features of 81 peptides, which are not enhancing infectivity (Table S5, Infection Rel EF-C $< 10\%$) [...] The independent variables (A) are all bioinformatic parameters and listed in the order of ascending P variable, Table S4. [...].

P16: The thresholds for classification were set as described in the following. Internal order was found if fibril formation is observed via TEM (Figure S33**Fehler! Verweisquelle konnte nicht gefunden werden.**) and characteristic β -sheet peak at 1630 cm^{-1} is observed via FTIR (Figure S34). Microscopic aggregation is positive for peptides showing count rate of scattered light above 100 kcps (Table S5) and [...]

P16, Figure S13, Caption: **A** Comparison of peptides' infection enhancement with and without N-terminal cysteine at conc. 1.3, 0.26 and 0 μM (Table S5). **B** Count Rate of scattered light of peptides with and without N-terminal cysteine (Table S5).

P18, Figure S15, Caption: [...] hydrophobicity (Fauchère, Table S4) and fulfill other prerequisites such as fibril formation and microscopic aggregation (Table S5).

P18, Figure S16, Caption: Peptides which are enhancing infectivity ($> 10\%$ relative to EF-C, 4-fold relative to virus only, Table S5), which do not form fibrils as analyzed by TEM (Figure S34).

P23: To find recurring patterns in active peptides, the sequences in the library (Table S3) [...]

P23, Figure S22, Caption: The coarse-grained amino acids were combined into all possible 7-mer patterns (16384) and matched with existing patterns in the peptide library (163, Table S3) were counted.

[...] To find amino acids mostly prevalent in high active peptides first the absolute abundance of amino acids in one category (low, medium, high) in the whole peptide library (163 sequences, Table S3) was determined.

P24, Figure S23, Caption: [...] **A** 8 of the peptides in the library have the pattern $P^+H^0P^+H^0P^0H^0P^0$ but do not enhance infectivity (Table S6). **B** Creating peptides based on the identified patterns $P^+H^0P^+H^0P^0H^0P^0$ and $P^+H^0P^0H^0P^0H^0P^0$ but mainly composed of amino acids prevalent in low active sequences result in weakly active peptides (Table S6). Fibril formation determined via TEM (Figure S34).

P25: [...] KIKIQINMWQRGD fibril formation and a positive zeta-potential (Figure S23 A, Table S6). [...] The only exception to this trend was observed for the peptide CRLSLSLS (medium active) (Table S6, Figure S23 B). [...] Remarkably, these non-active sequences show on average also lower sequence hydrophobicity compared to the positive examples (Figure 6B, Table S6).

Question 3. In Figure 4A, please provide all the eight kinds of possible combinations of physicochemical features. This is the root for the subsequent data mining section. Then, Figure 4 can be separated into two figures, not so large as the current version.

Answer 3. As suggested, we shifted the contents of previous Figure S11 to Figure 4 and separated previous Figure 4 in the new Figure 4 and the new Figure 5.

Consequently, we have adjusted all figure enumerations in the manuscript and SI.

The new Figure 4 and Figure 5 are shown below:

Figure 4 Evaluation of global property–activity relationship for infectivity enhancement with selected peptides as case studies to gain insight into the mode of action. Internal order is evaluated by TEM (Figure S33, Table S5, fibril formation) and FT-IR (Figure S34, Table S5 β -sheet content), surface charge by zeta-potential and infectivity is shown in absolute infection rates of peptides at 1.3, 0.26 and 0 μ M (Table S5). **A** Overview of all possible combinations of physicochemical features, which correlate most strongly with infectivity enhancement with examples for each possible combination. **1** The majority of peptides (75%), which enhance infectivity have high internal order, aggregate into μ m-sized aggregates, and have a positive zeta-potential, as shown exemplarily for the peptide CKFKFQF. **2** Without microscopic aggregation there is no infectivity enhancement even though other prerequisites are fulfilled such as fibril formation and positive zeta-potential shown for KFKFQFF. **3** Peptide fibrils which aggregate to clusters but have a negative surface charge are not enhancing viral infection, shown for CSISIQI. However, if the peptides are highly hydrophobic slight infection enhancement can be observed (e.g. for EIEIQINMWQ, Figure S14). Peptides which do not fulfill at least two of the features (assembly, aggregation, charge) cannot enhance infectivity as exemplarily shown for **4** (CHLHLQL or CKFKFQF with EGCG, **5** CKFKQFF **6** EIEIQINM, **7** KAKAQA. **8*** Peptides with a high amount of hydrophobic amino acids (W, Y) can enhance transduction although they are not assembling into fibrils if they form β -sheet rich, positively charged aggregates e.g. shown for HHHHKIKIKIYYYY. **B** Comparison of physicochemical properties and activity of CKIKIQI and CKIKQII. **i** TEM micrographs and β -sheet content determined by FT-IR (Figure S34), scale bar 1 μ m. **ii** Fluorescence microscopy of ThT-stained peptide fibrils with their respective light-scattering count rate (Table S5), scale bar 200 μ m. **iii** Zeta-potential **iv** Absolute infection rates of peptides at 1.3, 0.26 and 0 μ M concentration. **v** SFG spectroscopy at the amide I and II region.

Figure 5 Schematic overview of proposed mode of action over multiple scales from molecular to microscopic level. **Molecular** - Amphiphilic peptides with certain hydrophobicity can form β -sheet rich structures. **Nanoscopic** - Depending on the hydrophobicity of the β -sheet structures, short-range ordered amorphous aggregates or long-range ordered fibrils can form and further aggregate into μ m-sized particles. **Microscopic** - β -sheet rich aggregates can bind viruses and interact with cellular membrane by electrostatic and hydrophobic interactions. Created with BioRender.com

Question 4. Actually, the zeta potential values for amyloid-like peptide nanofibers vary in a relatively large range. The standard deviations for zeta potential values are not shown in the current dataset. Especially in SI document, some of the standard deviations of infections are shown but some are not. This will leave doubts to the related research fellows.

Answer 4. We thank the reviewer for emphasizing this important point. The standard deviations of the Zeta-Potential and the infectivity are summarized in **Table S5**. To address possible doubts, we now graphically show the standard deviations whenever possible, which includes **Figure 2**, **Figure 6**, **Figure 7**, **Figure S6**, **Figure S7**, **Figure S10**, **Figure S15**, and **Figure S30**

Additionally, we added **Figure S32**, the column “Std Dev. CR” in **Table S5**, **Table S6**, **Table S7**, and **Table S8** which summarize the experimental measurements with standard deviation for the peptides in the library, found via pattern analysis, data base screened amyloidal sequences and for aggregate size analysis, respectively.

The new figures and changes are listed below:

Manuscript, **P8, Figure 2C and D**: The contents of former Figure S5 A and B have been shifted to Figure 2C and D, respectively. Consequently, we have adjusted all Figure enumerations in the SI.

Manuscript, **P8, Figure 2E and 2F**: The aggregate size analysis was repeated to evaluate the number and aggregate size with standard deviations from triplicate measurements. The new data on aggregate number is inserted in **Figure 2E**, additionally we included the standard deviations of the count rate. Furthermore, we have exchanged the former plot in Figure 2F with the new data from aggregate analysis in **Figure 2F**. In the new plot we show the Count Rate correlating with number of aggregates in a logarithmic scale. An overview of the original data from aggregate analysis with standard deviations is listed in the newly added **Table S8**.

Furthermore, we included a short discussion on other correlations from the aggregate analysis in **SI Section 4.3, Figure S8** and would kindly refer to question 5 of reviewer #1 for more information on this.

Manuscript, P8, Figure 2

Figure 2 Correlation of experimental parameters with Log_{10} Infection Rel EF-C of a library containing 163 peptides. **A** Histogram of Log_{10} Infection Rel EF-C showing fibril forming (red) and non-fibril forming (blue) peptide distribution analyzed via TEM (Figure S33). **B** Histogram of Log_{10} Infection Rel EF-C showing ThT-active (red) and ThT-inactive (blue) peptide distribution (Table S5). **C** Zeta-potential plotted against Log_{10} Infection Rel EF-C. Error bars indicate standard deviation from triplicate measurements (Table S5). Linear fit with Pearson correlation factor $R = 0.67$. **D** Count rate of scattered light plotted against Log_{10} Infection Rel EF-C. Linear fit with Pearson correlation factor $R = 0.65$. Error bars indicate standard deviation from triplicate measurements (Table S5). **E** Selection of peptides with high and low count rate showing relationship between number of particles and count rate of scattered light (Table S8). Peptide fibrils were preincubated at 1 mg/mL one day and diluted to 0.1 mg/mL with ThT solution (50 μM) just before measurement (Figure S32). Number of aggregated fibrils ($>10 \mu\text{m}^2$) were quantified via fluorescence microscopy in an area of $1330 \mu\text{m} \times 1330 \mu\text{m}$ in triplicate measurements via automated counting (3D object counter ImageJ), scale bar 200 μm . **F** Log_{10} Count rate from zeta-potential measurements of representatively selected peptides plotted against averaged number of aggregated particles in an area of $1330 \mu\text{m} \times 1330 \mu\text{m}$ (Table S8). Error bars indicate standard deviation from triplicate measurements, linear fit (solid line) demonstrates a high correlation with a Pearson R of 0.84.

In former **Figure 5C**, now **Figure 6C**, error bars for Zeta-Potential and Infection Rates were included.

Manuscript, P15, **Figure 6**

Figure 6 Preserved pattern in infectivity enhancing peptides can be applied to create new active sequences confirming property-activity relationship found for original library. **A** Coarse graining infectivity enhancement in low (< 10%) medium (10-70%) and high (> 70%, Infection Rel EF-C) and coarse graining amino acids according to their hydrophilicity (H = hydrophobic; P = hydrophilic) and side chain charge at pH 7.4 (+ = positive, - = negative, 0 = neutral) enables identification of most abundant sequence pattern in highly active peptides (**Table S5**) that is an alternating amphiphilic motif $P^+H^0P^+H^0P^+H^0P^+$ and $P^+H^0P^0H^0P^0H^0P^0$. **B** New peptides are created by combining the preserved pattern $P^+H^0P^+H^0P^0H^0P^0$ with the identified best-performing amino acids. 8 of 10 newly created peptides are enhancing infectivity (> 10% Infection Rel EF-C at 1.3 μ M, **Table S6**) and overall confirm the identified property-activity relationship. **C** Graphical visualization of the most important property-activity relationship zeta-potential plotted versus Log₁₀ Infection relative to EF-C. The color code represents the Log₁₀ Count Rate from light scattering, which can be traced back to microscopic aggregation (**Table S6**). **Error bars indicate standard deviation from triplicate measurements.** **D** Selected representative example for fibril morphology (TEM, **Figure S25**, scale bar 1 μ m) and cell interaction (confocal microscopy, scale bar 50 μ m) of an active peptide (CKIWIQIY), which is based on the preserved pattern. Confocal image shows merged measurement from transmission, Proteostat (red, fibrils) and Hoechst (blue, cell nuclei) channels. **E** Selected representative example for amorphous morphology (TEM, scale bar 1 μ m) and no cell interaction (confocal microscopy, scale bar 50 μ m) of an inactive peptide (RINFWFW), which is based on the preserved pattern. Confocal image shows merged measurement from transmission, Proteostat (red, fibrils) and Hoechst (blue, cell nuclei) channels. CKIWIQIY and RINFWFW were both susceptible for amyloid sensitive dye staining (**Figure S24**), but RINFWFW does not attach to cells in accordance with its low activity.

In former **Figure 6B**, now **Figure 7B**: error bars for Zeta-Potential and Infection Rates were included.

Manuscript, P16, **Figure 7**

Figure 7 Infectivity enhancing properties of selected amyloid peptides from pathogenic and functional origin. **A** Infection rates of EF-C, and peptides ISKLEYSNFSVRY, LANWMCLAKW, VHDCVNITIK at 6.5, 1.3, 0.26, 0 μM concentration. Numbers at the end of bars indicate *n*-fold infectivity enhancement compared to virus-only infectivity (0 μM) (Table S7). **B** Zeta-potential plotted against Log₁₀ Infection Rel EF-C. Color indicates Log₁₀ Count Rate by scattered light. Three peptides (box) that show identified important physicochemical properties that have positive zeta-potential, show fibril formation, and aggregation are enhancing infectivity (Table S7). Error bars indicate standard deviation from triplicate measurements. **C-E** TEM micrograph (1 mg/mL, PBS, Figure S36, S37 scale bar 100 nm) and confocal microscopy (scale bar 20 μm) of acyl phosphatase derived peptide ISKLEYSNFSVRY (**C**), human lysozyme derived peptide LANWMCLAKW (**D**) and human prion protein derived peptide VHDCVNITIK (**E**). 20 μg/mL preassembled peptides (stained with Proteostat, red) were incubated with *Henrietta Lacks* (HeLa) cells (nucleus stained with Hoechst, blue). For single channels see Figure S28.

New added **Figure S32** shows representative widefield microscopy images.

SI, P51, **FigureS32**

Figure S32 Representative selection of widefield microscopy images of peptides from the peptide library. Images with a dark background (white scale bar) are fluorescence microscopy images, and images with a bright background (black scale bar) are brightfield microscopy images. Peptides were diluted from 10 mg/mL in DMSO to 1 mg/mL in PBS and incubated for 1d at RT. Before the measurements they were diluted to 0.1 mg/mL with 50 μ M ThT-solution in PBS, scale bar 200 μ m.

SI, P10, Figure S6

Figure S6 Graphical representation of calculated hydrophobicity (Table S4) plotted versus \log_{10} Infection Rel. EF-C (Table S5). The color code represents fibril formation (red) and no fibril formation (blue) as observed via TEM (Figure S33, Table S5). Interestingly, there are no active peptides (dashed line, Infection Rel EF-C <10% or \log_{10} Infection Rel EF-C <-1) with a hydrophobicity lower than 0.17. From these active peptides 90% are forming fibrils. The majority (69%) of peptides with a hydrophobicity higher than 0.17 are also forming fibrils, illustrating a positive correlation between hydrophobicity, fibril formation and infectivity enhancement.

SI, P10, Figure S7

Figure S7 Characterization of β -sheet and monomer to assembly conversion of the peptide library (Figure S34, Table S5). **A** β -sheet content obtained from attenuated total reflection (ATR)-FTIR measurements plotted against \log_{10} infection relative to EF-C. **B** Conversion of monomers to assembled structure contain data of 80 peptides plotted against \log_{10} infection relative to EF-C.

SI, P44, Table S5: The column “Std. Dev. CR” have been added.

SI, P13, Figure S10

Figure S10 Multiparameter analysis of assembly (Figure S33), secondary structure (Figure S34), surface charge and microscopic aggregation (Table S5). **A** β -sheet content plotted against Log_{10} Infection Rel. EF-C. The color bar indicates the zeta-potential. **B** Peptides with a high β -sheet content, which are microscopically aggregating (high count rate) and have a positive zeta-potential are enhancers of viral transduction (top right corner). The color bar indicates Log_{10} Infection Rel. EF-C. **C** Conversion of monomers to assembled structures (Conversion Rate, CR) plotted against Log_{10} Infection Rel. EF-C of 80 selected peptides (Table S5). The color bar indicates the zeta-potential. Note, that for one peptide (CKFKQFF) with -2.2 Log_{10} Infection Rel to EF-C shows no infectivity enhancement despite its positive zeta-potential and high CR. For this peptide the turbidity is very low and amorphous aggregates rather than fibril formation is observed, which indicates no microscopic aggregation. **D** Multiple linear regression (constrained linear least-squares problems solution) with the equation: $y = A_0 + A_1 \cdot x_1 + A_2 \cdot x_2 + \dots + A_5 \cdot x_5$ and zeta-pot, Log_{10} Count Rate, Fibril formation, H_{fauchere} , ThT-activity and β -sheet as independent variables (A, listed in the order of ascending P variable) and Log_{10} Infection Relative EF-C as dependent target variable (y). Linear Fit with Pearson correlation factor 0.82. Statistically significant parameters ($P < 0.05$) are highlighted. **E** Multiple linear regression (constrained linear least-squares problems solution) with the equation: $y = A_0 + A_1 \cdot x_1 + A_2 \cdot x_2 + \dots + A_7 \cdot x_7$ and Fibril formation, zeta-pot, Log_{10} Count Rate, β -sheet, H_{fauchere} , ThT-activity and Conversion Rate as independent variables (A, listed in the order of ascending P variable) and Log_{10} Infection Relative EF-C as dependent target variable (y) with 79 peptides from the library which include CR as well as FTIR data. Linear Fit with Pearson correlation factor 0.91. Statistically significant parameters ($P < 0.05$) are highlighted.

SI, P18, Figure S15

Figure S15 Peptides with negative zeta-potential (marked) slightly enhance infectivity if they have high (>0.6) calculated hydrophobicity (Fauchère, Table S4) and fulfill other prerequisites such as fibril formation and microscopic aggregation (Table S5).

SI, P33, Figure S30

Figure S30 β -sheet content plotted against Log Infection Rel EF-C of 24 literature reported peptides. Color scheme represents zeta-potential. Left: All peptides regardless of the count rate are shown. Right: Peptides above a critical count rate of 100 kcps is shown. All tested peptides with a high β -sheet content and a positive zeta-potential are enhancers of viral transduction (top right corner).

Question 5. Most important, the data mining is highlighted at the beginning of the title. However, the data mining procedure and evaluation are not evidently provided in the whole article. Please display the data mining procedure in both abstract, methods, results, discussion, and conclusion.

Answer 5. We thank the reviewer for giving us the opportunity to improve our manuscript. As suggested, we have added the following passages to each respective section of the manuscript:

Manuscript, abstract, **P2:** [...] Data-mining is an efficient method to systematically study structure–function relationship and unveil patterns in a database. [...]

Manuscript, results, **P6:** To unveil the structure–property–activity relationship of infectivity enhancing peptides, we performed a data-mining approach. This procedure involves data-generation by creating a peptide library, processing and analyses of the peptide measurements, establishing a model for the structure–property–activity relationship, and finally interpretation and prediction based on the established relationship.

Manuscript, discussion, **P18:** [...] To test this, we applied a data-mining approach that includes the creation of a suitable dataset, the evaluation of corresponding data, and interpretation of found relationships. [...]

Manuscript, conclusion, **P21:** By applying a data-mining approach, we could perform a multi-parameter regression and unveil crucial co-dependencies in the peptide properties that govern their activity and act on length scales from nano to micrometers [...].

Manuscript, methods, **P21: Data-Mining.** The herein applied data-mining procedure covers data-generation, processing, data extraction and evaluation, model development, prediction, and interpretation to evaluate the identified property–activity relationship.

The first step involved **data-generation**, where 163 distinct peptides based on an infectivity-enhancing peptide EF-C (QCKIKQIINMWQ) were synthesized (**Table S2**), six physicochemical and one biological properties were measured (see **SI Section 1.1** and method section below). The collected data was then **processed**, as summarized in **Table 1**. Twelve bioinformatic properties including peptide net charge, hydrophobicity, isoelectric point, aliphatic index, hydrophobic moment index, boman index, instability index, graph shape index, upsilon steric parameter, smoothed upsilon steric parameter, polarizability, and normalized van der Waals volume were calculated via the “peptides” package in R (see **SI Section 1.2**).⁴⁰

Table 1 Summary of pre-processing methods to obtain output data for data-mining from the readout data of experimental descriptors.

Descriptor	Method	Readout	Pre-processing	Output
Infectivity enhancement	β -galactosidase assay	β -galactosidase luminescence (infection rate intensity) in relative light units per second (RLU / s)	Calculation of n-fold infection in presence of peptides relative to virus only infection in absence of peptides. Calculation of ratio between n-fold infection of peptides relative to n-fold infection of EF-C (Infection Rel. EF-C).	Logarithmic infectivity relative to EF-C (Log_{10} Rel Infect EF-C)
Fibril formation	TEM	Micrographs	Qualitative evaluation of high aspect ratio nanostructures (fibrils)	Fibril formation observed (1) or not observed (0)
Monomer to nanostructure conversion	Conversion Rate	Fluorescamine fluorescence intensity	Fluorescence intensity ratio before and after filtration	Percentual fluorescence intensity ratio (conversion rate)
Amyloid structure	ThT-fluorescence	Thioflavin-T fluorescence intensity	N-fold fluorescence intensity of peptides compared to buffer control sample	ThT-active (N-fold >2), or ThT-inactive (N-Fold < 2).
β -sheet content	FT-IR	Infrared absorbance (1300 and 1800 cm^{-1}) of peptide samples	Data-smoothing, normalization, calculation of second derivate of absorbance intensity, calculation of peak integral ratios between peaks at $1625\text{-}1635\text{ cm}^{-1}$ (β -sheet) and approx. $1650\text{-}1700\text{ cm}^{-1}$ (native structure).	Percentual β -sheet content
Charge	Zeta-Potential	Zeta-Potential in mV	-	Zeta-Potential in mV
Microscopic aggregation	Light Scattering Count Rate	Light Scattering Count Rate in kilo counts per second (kcps)	-	Light Scattering Count Rate in kilo counts per second (kcps)
Microscopic aggregation	Widefield Microscopy	Microscopy images	Count area size and number of aggregates ($>10\text{ }\mu\text{m}^2$) via 3D objects counter (ImageJ)	Aggregate area size and number in a microscopy image.

The next step was **extraction** and **evaluation** of the collected data: For an initial screening every descriptor was summarized in a table and imported to the data-mining software Orange3 (Orange3-v.3.34.0, Anaconda) to find the best correlating descriptor combinations via the functions “scatter plot \rightarrow find informative projections” and “pearson correlations”. The Pearson correlation coefficients for every descriptor were

calculated via the function “correlation plot” of the software OriginPro 2021 v.9.8.0.200 (OriginLab). The property–activity relationship was identified using the function “constraint multiple regression” analysis (constrained linear least-squares problems solution with the equation: $y = A_0 + A_1 \cdot x_1 + A_2 \cdot x_2 + [\dots] + A_5 \cdot x_5$) of the software OriginPro 2021 v.9.8.0.200 (OriginLab). The model was evaluated using various statistical metrics such as the Pearson correlation coefficient and the statistically significant descriptors were identified by $P < |t|$, $t = 0.05$.

The established linear regression model was **validated** with *de novo* peptides **predicted** from a pattern analysis technique and with existing peptides from a database screening.

The pattern analysis aims to find amino acid patterns and amino acids that are prevalent in high-active peptides (see **SI Section 7**). Briefly, amino acids were first classified into four categories based on their hydrophilicity and charge at pH 7.4 using the Kyte–Doolittle hydropathy scale.⁸⁴ The infectivity enhancement was categorized in low (< 4-fold virus only, < 10% relative to EF-C), medium (10–70% relative to EF-C), high (> 70% relative to EF-C). All possible 7-mer patterns were then generated from these coarse-grained amino acids and matched against existing patterns in the peptide library. To identify patterns that are more prevalent in high-active sequences compared to medium- or low-active sequences, an activity index of a pattern p ($AI(p)$, **eq I**) was calculated by subtracting the sum of its relative numbers in medium- and low-active sequences from its relative number in high-active sequences as in:

$$AI(p) = \frac{N_{\text{high}}(p)}{\sum N_{\text{high}}} - \frac{N_{\text{medium}}(p)}{\sum N_{\text{medium}}} - \frac{N_{\text{low}}(p)}{\sum N_{\text{low}}}. \quad \text{eq I}$$

For the amino acid analysis, the absolute abundance of amino acids in one category (low, medium, high) in the whole peptide library was first determined. The relative abundance of amino acids “*Rel AA*” was then calculated by determining the number “ N ” of peptides containing a certain amino acid “*AA*” relative to the total amount of peptides of one category (**eq II**).

$$Rel\ AA = \frac{N(\text{peptides containing } AA)_{\text{category}}}{\sum \text{peptide}_{\text{category}}} \quad \text{eq II}$$

All pattern and amino acid analyses were conducted using Python scripts.

The prediction and evaluation of new peptides were conducted by retranslating the coarse-grained best-performing patterns with best-performing amino acids and worst-performing amino acids to yield 10 peptide sequences designed to be active and 8 peptides designed to be inactive. To avoid human bias, the selection of amino acids from the pool of best-performing and worst-performing ones was conducted randomly.

For the data-base screening 538 short (< 20 amino acids) peptides across various sources (176 peptides from PDB⁸⁵, 260 peptides from a report⁵¹ and 102 peptides from Waltz database⁸⁶) were collected by their bioinformatic properties (hydrophobicity, isoelectric point, net charge, calculated the “peptides” package in R⁴⁰) and by any reported information on structure formation (TEM, FT-IR and ThT-fluorescence). Ultimately, 24 different peptides (**SI Section 8**) were selected based on fibril formation by TEM and preferably by net charge (> 0), hydrophobicity (> 0.2), β -sheet formation by ThT fluorescence.

Question 6. Please check all the abbreviations throughout the manuscript. For example, "CAR T-cell therapy", "HIV", "HIV-1", "TZM-bl cells", etc. The full name should be presented at the first time. The SI document is suggested to check through.

Answer 6. We thank the reviewer for giving attention to minor errors, which were all corrected in the manuscript. We should note, that there is no full name for TZM-bl cells.

In the manuscript:

P3 chimeric antigen receptor (CAR), **P3** human immunodeficiency virus 1 (HIV-1), **P3** glycoprotein 120 (gp120), **P9** hydrophobicity (H_{fauchere} , **P11** phenylalanine (F) or isoleucine (I) amino acids with alanine (A), **P16** Henrietta Lacks (HeLa) cells, **P18** semen-derived enhancer of viral infection (SEVI)

In the SI:

P3 Transmission electron microscopy (TEM), **P3** phenylalanine (F), tryptophane (W) and tyrosine (Y), **P5** alanine (A), isoleucine (I), leucine (L) and valine (V), **P9** fourier transform infrared spectroscopy (FT-IR), **P10** attenuated total reflection (ATR), **P24** H^0 hydrophobic non-charged, P^0 hydrophobic non-charged, P^- hydrophilic negatively charged, P^+ hydrophilic positively charged

Reviewer #3 (Remarks to the Author):

The present manuscript provides a comprehensive analysis of a range of small peptides on their ability to enhance viral infectivity. This property could be of use for viral uptake related applications. Correlations are obtained for various parameters tested and it is concluded that a combination of factors must be met for efficient enhancement of viral infectivity. Some of these factors were known already, such as the need for positive charges. One main new discovery is that the size of the aggregates matters, with larger aggregates being more potent. This information might well prove useful for understanding the effect of amyloids as adjuvants and should be of value to the field of viral enhancement by aggregates.

We thank the reviewer for a careful reading and positive evaluation our manuscript.

Question 1. The novelty to the amyloid field as a whole is rather limited. Much emphasis has been placed on correlations of activity with hydrophobicity and other physical parameters. While solid insights are obtained, they are not necessarily novel. That is, extensive work by Dobson and many others has established nearly the same concepts for amyloid protein misfolding in a series of studies over 2 decades ago. This work does not appear to be cited and it diminishes the broader significance of the present study.

Answer 1. We thank the reviewer for giving us the opportunity to clarify this important aspect. We are aware of the excellent work by Dobson and others who pioneered research on amyloid aggregation and we have now included several relevant articles by Dobson and other authors. Concerning the novelty of our findings in view of these seminal papers, we would like to highlight that most studies in the amyloid misfolding field focus on angstrom–nm-sized aggregation phenomena (amyloid oligo and protofilament formation)^{9–12} and rarely investigate the molecular features giving rise to μ -sized aggregation.

Even though few reports discuss μ -sized fiber-fiber aggregation, these studies mainly focus on

- simulations to explain how different fiber lengths of amyloid A β affect the stiffness of μ -sized fiber-fiber aggregated “plaques” through a balance between elastic deformation and “adhesive” forces, which are not further defined.¹³
- experiments which show fiber-fiber lateral interaction for nm- μ m sized highly ordered alignments,^{14–16} but are not investigating fiber-fiber lateral interaction to μ -sized particles composed of fibrils which aggregate with each other in a non-ordered fashion.
- examples where fibrils show μ -sized aggregation in a highly ordered fashion likely due to hydrophobic and VdW interactions,^{17,18} but they are not comparing them to fibrils where μ -sized aggregation is not observed, thus not deconvoluting the missing part between fibril formation and fibril-fibril attraction.
- time-dependent slow (hours-days) but highly ordered fibril aggregation phenomena of proteins across length scales,^{19–21} which are likely following an association pathway controlled by seeding and nucleation events of

monomers to filamentous fibrils, than lateral fiber-fiber association of rapidly (seconds) matured assemblies which is the proposed interaction mechanism here.

Noteworthy, most of these studies on μm -sized aggregation are conducted with oligopeptides or complex folded proteins and not with simple, short (< 20 amino acids) synthetic, self-assembling peptides. To the best of our knowledge, studies dealing with synthetic short peptides for μm -sized aggregation²² do not systematically investigate the molecular driving forces.

Thus, short self-assembling peptides offer the advantage of deconvoluting molecular sequence contributions more effectively (e.g. shown for the examples CKIKQII and CKIKIQI (**Figure 4**) or CKIKIQI and KIKIQIC (**Figure S14**). Our study provides fundamental structural features for retroviral transduction enhancement and we clearly show that μm -sized aggregation is essential. We believe that our results are of great importance for scientists interested in gene delivery, nevertheless, our findings are still relevant for the amyloid protein misfolding community. To our understanding, it is very challenging to obtain correlations of amyloid-forming proteins from the molecular to the mesoscale. Here, our study provides information of the impact of single amino acids on amyloid structure formation and mesoscale aggregation.

We have now included a section in the discussion where we provide a broader context with respect to the amyloid protein misfolding:

Manuscript, **P20**: Pioneering works have put much efforts in identifying amyloid fiber evolution mechanisms on a molecular and nm-size level, e.g. for oligo and protofilament formation.⁷²⁻⁷⁴ On the inter-fiber mesoscale lateral aggregation was traced back H-bonds⁷⁵ and VdW polar interaction modes⁷⁶ and to hydrophobic interactions.^{77,78} Interestingly, experiments and simulations showed that dynamic, flexible fibrils are less prone to align into multi-fiber strands.^{37,79} However, the [...]

Question 2. There are also numerous proven algorithms available that have good predictive qualities for aggregation and amyloid formation, making the present study less novel. One of the parameters that has also been considered in the literature, but which is not discussed here, is beta-sheet propensity, which leads to a preference for beta-branched side chains in many amyloids. This is only a minor concern, but it was somewhat surprising that it was not included.

Answer 2. We thank the reviewer for pointing this out and would like to take this opportunity to clarify that besides the intermolecular aggregation, which can give rise to beta-sheet secondary order, we here consider aggregation on the micrometer scale, which involves fibril-fibril association. Importantly, we must differentiate that a beta-sheet forming fibril is not necessarily aggregating into micrometer large structures, as observed in this study for example for the non μm -sized aggregating, beta-sheet rich peptide fibril IKFLSVN (Figure 7, Figure S37, Figure S39, Table S7). While the molecular structure can be used for predicting the propensity to form β -sheet rich structures, e.g. with algorithms such as PASTA 2.0, Waltz, APPNN, PATH, to the best of our knowledge there is no algorithm to predict the multi-scale aggregation from the molecular sequence. Since the evaluation of the reliability of these algorithms to predict fibril formation and μm -sized aggregation is beyond the scope of this study, we do not

discuss these algorithms here, however we would like to kindly refer the reviewer to a recent report by us, which focused exactly on this question.⁷

Interestingly, by analyzing the amino acid patterns in our library, we have found that certain amino acids known to promote β -sheet fibril formation, are predominantly present in peptides that are infectivity enhancing and form μm -sized aggregates. This connection between β -sheet prone amino acids has not been previously reported for aggregation in the micrometer size, which is why we now included a short paragraph on this.

Manuscript, **P15**: Noteworthy, we observed that most of the herein found amino acids from highly active sequences and weakly active sequences have been known to promote and break β -sheet fibril formation, respectively (**SI Section 7**).

SI, **P25**: By applying the amino acid analysis (**Figure S22 B**) we find the amino acids W, M, N, I, C, Q, K, F and Y present in a higher frequency in highly active sequences “active amino acids” compared to amino acids S, G, R, H, T, V, P, A and L which are predominantly present in weakly active peptides “inactive amino acids” (**Figure S22 B**). Interestingly, most of the “active amino acids” W, M, I, C, Q, F and Y are known to promote β -sheet amyloid fibril formation except of N, K (β -sheet breaking), whereas the “inactive amino acids” are either neutral (R, A, D, G) or β -sheet breaking (E, H, P, S) except of T, L, V (β -sheet promoting).^{30,31}

Question 3. The paper categorizes aggregates into fibril forming and amorphous aggregate forming peptides. It should be noted that this cannot be correlated merely with intrinsic properties of peptide sequence alone and it is also difficult to decipher simply based on one experiment under one condition. The problem is that nearly all amyloids can form fibrils or more amorphous material, it just matters on the conditions. Sometimes they can co-exist on the same EM grid and sometimes only subtle changes in conditions can switch from one to the other. Sometimes the behavior can even be complicated by the exact purification protocol prior to dissolution. The detection can also be difficult in some cases. The image shown for amorphous aggregates could indeed be fully devoid of fibrils, but it could also contain lots of fibrils.

Answer 3. We thank the reviewer for drawing attention to the topic of polymorphism in amyloid peptides. We agree with the reviewer that amyloid peptides are often observed in different states also within the same EM-grid. In our study we evaluated each EM-grid carefully and entirely to ensure not to miss a certain nanostructure type. Peptides which showed exceptional behaviour, such as infectivity-enhancing peptides, which do not form fibrils (**Figure S16, Table S5**) have been prepared and measured at least in duplicates. Each EM-micrograph (**Figure S33**) was selected representatively, showing predominantly fibrillar structures, a mixture of fibrils and amorphous aggregates or only amorphous aggregates depending on what nanostructure-type was predominantly observed during the analysis for the entire grid.

Specifically, a peptide was defined as “amorphous aggregate” if no fibrillar structure was observed but only amorphous aggregates. In contrast, peptides defined as “fibrillar” may also include amorphous structures.

To clarify these qualitative evaluation criteria for the reader, we now included the following paragraph in the

SI Section 1.1, **P3**: A peptide was defined as non-fibrillar if no fibrillar structure was observed once on the entire TEM-grid but only amorphous aggregates. In contrast, peptides defined as fibrillar may also include amorphous structures.

Since the analysis of different preparation protocols for the entire library is beyond the scope of this report, we now added a short discussion on the limitations of TEM as a method to qualitatively evaluate fibril morphology:

Manuscript, **P18**: For example, TEM is a qualitative high-resolution method for structural characterization on a nanometer scale, but it cannot quantify reliably fibril polymorphism and it lacks information on intermolecular interactions, [...]

Question 4. Sometimes fibrils can appear after stripping less ordered proteins or peptide from their surface. Surface binding of non-fibrillar material can make ThT fluorescence highly problematic. The authors need to acknowledge this potential complication more clearly.

Answer 4. We agree that ThT fluorescence can be subject to various fluctuations not related to the actual structure. It is well reported in literature, that ThT fluorescence can be observed for structures not forming beta-sheet structures simply because of structural polymorphism,²³ electrostatic interactions with positive charged particles,²⁴ and in the presence of aromatic amino acid such as phenylalanine, tryptophane and tyrosine.^{25,26} To clarify the limitations of ThT-fluorescence as a methodology to determine β -sheet structures we included the following section in the manuscript discussion part and refer to the SI on P3 for more details:

Manuscript **P18**: To detect the formation of β -sheet rich fibrils in bulk, we utilized the amyloid-sensitive ThT dye. However, it should be noted that ThT-fluorescence has been associated with false-positive results (**SI Section 1.1**). All these [...]

Question 5. The analysis of small peptide fragments from naturally occurring pathogenic amyloids is useful in the sense that it shows how their model has predictive power. Unfortunately, the authors go beyond this point, which leads to several concerns. First, the study uses small peptide fragments only, which makes it harder to learn much about the actual pathological proteins or peptides. Second, there is little evidence that these amyloids act by being adjuvants in vivo. In fact, it seems that nearly all peptides are not very potent at promoting infection based on the data obtained.

Answer 5. We thank the reviewer for raising these concerns. Our intention of using pathogenic and amyloid peptides from various backgrounds was to validate the

general properties for infectivity enhancement beyond our library. We must clarify that we did not intend to claim that the original proteins can be used as *in vivo* adjuvants. The evaluation of proteins is beyond the scope of the present study. However, exemplarily, we tested the enzyme lysozyme for its zeta-potential and μ -sized aggregation behavior (**Figure S40**) and found that at the tested conditions, the enzyme shows largely different physicochemical properties than the corresponding peptide fragment LANWMCLAKW, which indicates that one cannot simply extrapolate from the peptide fragments to the source protein.

To clarify this for the reader, we included a comment regarding transferability from short segments to proteins aggregation and **Figure S40**:

Manuscript, **P19**: It is important to note that while these protein fragments can enhance infectivity, their physicochemical properties cannot be simply extrapolated to the parent protein, as shown exemplary for lysozyme and LANWMCLAKW (**Figure S40**).

SI, **P68**, **Figure S40**.

Figure S40 Comparison of physicochemical properties of the full protein lysozyme and a sequence fragment Lysozyme₂₅₋₃₄ LANWMCLAKW. **A** No μ -sized aggregation or ThT-fluorescence can be observed for lysozyme, while LANWMCLAKW is yielding clearly visible μ -sized aggregates in brightfield microscopy and ThT-fluorescent aggregates via fluorescence microscopy, scale bar 200 μ m. Preparation of both samples at 809.7 μ M (1 mg/mL LANWMCLAKW) in PBS (10 vol% DMSO), incubation for 1d at rt. Just before microscopy measurement samples were diluted to 80.9 μ M with 50 μ M ThT-dye solution in PBS. **B** Zeta-Potential for lysozyme and LANWMCLAKW and **C** derived Count Rate of scattered light for lysozyme and LANWMCLAKW from zeta-potential measurements. Error bars indicate Std. Dev. from triplicate measurements. Samples were incubated at 809.7 μ M in PBS (10 vol% DMSO), for 1d at rt before diluting for zeta-potential measurement samples to 80.9 μ M with 1 mM KCl.

Question 6. Third, it is argued that the approach of looking for micron-sized aggregates of amyloid fibrils is often ignored in the field of pathogenic amyloids and that studying intrinsic properties of peptides (as done in the present study) could be helpful. I am not so convinced by this statement, as many studies have extensively looked at the correlation between amyloid size and toxicity. In the vast majority of cases, it has been concluded that the large, micron-sized aggregates are rather inert (so the exact opposite of the present study) and that smaller oligomers or protofibrils are more toxic.

Answer 6. As discussed in more detail the first answer, to the best of our knowledge studies focusing on amyloid aggregation size evaluate nm-size regimes, *i.e.* fibril formation^{9–11} or μm -scaled fibril alignment^{13–22} but not μm -sized aggregates composed of unordered fibrils.

We have now included an improved definition of μm -sized aggregation as it can be confused with the highly ordered μm -sized crystalline type fibers.

Manuscript, **P6:** In this study, we use the term microscopic aggregation to describe the fiber-fiber interaction into less ordered micrometer-sized particles and exclude the highly ordered μm -sized crystalline alignments that have been previously studied for amyloid aggregation.^{36–38}

Analogously, toxicity studies refer to fibril formation and not on μm -sized fibril aggregation.^{27,28} A hypothesis is that these toxic amyloids form oligomeric species because they aggregate slowly in the timeframe of days. In contrast our herein reported peptide fibrils are not toxic and are forming fibrils and μm -sized aggregates within seconds (see newly added **Figure S4** and **Figure S9** and corresponding answer to question 1 of reviewer #1). These observations support the hypotheses that fibrillation kinetics may be responsible for high oligomeric species, which cause toxicity.

Another current hypothesis is that the end caps of amyloid fibrils induce toxicity by disturbing (“pinching”) the cellular membranes.²⁹ Amyloid fibrils which have a high lateral aggregation have less exposed terminal ends and are less toxic.²⁹ This is well aligned with findings from our study where we observe no toxicity for large aggregates but rather show good cell-viability for HeLa cells (**Figure S2**) and in other reports with similar peptides for other types of cells.^{30–32}

Importantly, we do not state here that μm -sized amyloid aggregates are toxic, but that molecular reasons for μm -sized aggregation is elusive and often not considered in the context for biological activity, for instance infectivity enhancement by virus-fibril-cell interactions.

To further provide broader context of our study on the amyloid field we included following sections:

Manuscript, **P3:** A hypothesis that distinguishes pathogenic from functional amyloids is based on the kinetics of their assembly process. It has been proposed that oligomeric species rather than mature, thermodynamically stable amyloid fibrils are related to cytotoxicity.^{16–18} Oligomers which prevail for a long time in solution can interact and also refold other monomeric proteins thus acting as multipliers for proteins misfolding and aggregation.^{19,20} In contrast, reports on functional amyloids typically show very fast

and quantitative assembly without long lag times, thus preventing the formation of cytotoxic oligomers.^{21,22}

Manuscript, **P20**: Amyloid fibrils that aggregate expose fewer terminal ends that are known to disrupt cellular membranes.⁶⁷ The fibrillar aggregates reported in this study do not show toxicity but good cell-viability for HeLa cells (**Figure S2**) as well as other cell types for similar fibrils, as reported elsewhere.^{68–70} Microscopic aggregation has been previously observed in the non-diseased state of endogenous semen protein prostatic acid phosphatase,⁷¹ which suggests a broader biological significance of micrometer-sized aggregation for amyloids beyond simple disease-related associations.

Question 7. In this context, it is also important to note that the same protein can often take on many different aggregation states from smaller oligomers, short fibrils, all the way to micron-sized bundled fibrils of different polymorphs. There clearly maturation factors at play here that allow the same sequence to take up many different conformations and aggregation state and many of those will depend on cellular context.

Answer 7. We agree with the reviewer on this point. To make the reader aware of this complexity, we include this consideration in the discussion.

Manuscript, **P17**: It is well-known that amyloid polymorphism can greatly influence its bioactivity.^{52,53} However, the systematic study of different fibril morphologies is outside the scope of this study.

Question 8. Another important practical consideration regarding the use of pathogenic amyloids as adjuvants would be the potential danger that these peptides might act as seeds and cause neurodegenerative or other diseases.

Answer 8. We thank the reviewer for raising this concern. We do not intend to use these peptides found from pathogenic origin as infectivity-enhancing adjuvants but report them to validate the universal property-activity correlation found from systematically studying the EF-C based library. To avoid misunderstandings, we added the following sentence on

Manuscript, **P19** However, it is not recommended to use peptides derived from the original pathogenic origin for therapeutic purposes because they may act as potential seeds for disease-related aggregation events.

Question 9. The discussion about the membrane interaction appears highly speculative. There is no structural information obtained for any of the peptides, and it is not clear which regions interact with the membrane. Much hydrophobic surface might be covered by inter filament interactions and is perhaps not even available for membrane interaction. Yes, the Wimley-White scales indicate that hydrophobic side chain have an easier time at partitioning into the membrane

and such residues can clearly promote membrane binding. Having said that some peptides, such as TAT, can promote cell entry through the use of positively charged clusters. I suggest making this section more convincing or take it out. It appears far too speculative as written.

Answer 9. We thank the reviewer for giving us the opportunity to improve the discussion on the fibril–membrane interaction. We have now included literature reports studying different hydrophobic patches on the surface of various amyloid fibrils for lateral fiber–fiber interactions to support our discussion. Recently, we could show that virus infectivity can be tuned by applying different hydrophobic patches on fibrils (preliminary, unpublished data of work in progress), therefore we argue that the membrane interaction mechanism via hydrophobic patches is reasonable, however as we do not have structural evidence, we made clear that this discussion is speculative.

Manuscript, **P19:** The interplay between polar and non-polar interactions, so-called hydrophobic patches, at the fibril surface can determine the fate of ordered or less ordered fiber–fiber aggregation across length-scales^{36,81} and might explain why some amyloid fibrils form unordered large ($> 10 \mu\text{m}^2$) aggregates while others stay isolated or form well-aligned bundles. Our data shows that regular arrangement of polar and non-polar amino acids facilitates stable β -sheet rich structures and SFG measurements indicate that hydrophobic surface interface dominates for microscopically aggregating peptides. Therefore, surface hydrophobicity likely is a key driving force for formation and μm -sized aggregation⁸² of amyloidal fibrils and an important characteristic besides charge–driven interactions to associate with viruses and cells.⁸³ The fewer water-accessible areas may result in higher fibril aggregation which may be at the same time the driving force for cell and virus interaction. However, structural evidence of these interactions is needed to support these hypotheses, which is beyond the scope of this study.

Other changes to the manuscript

We have adjusted all Figure and Table enumerations and highlighted the changes.

Further changes in the manuscript:

P4: transmission electron microscopy (TEM)

P7: [...] our data shows that the logarithmic number of μm -sized aggregated fibrils determined by fluorescence microscopy (**Figure 2E**, **Figure S32**) correlates linearly with the logarithmic scattering intensity [...]

P12: Thereby, we conclude that multiple conditions including (I) a positive surface charge, (II) ordered structure, (III) microscopic aggregation are required for highly efficient retroviral transduction enhancement (**Figure 5**).

P21: The specific factors that determine the ordered or unordered aggregation of micrometer-sized particles are currently unknown, but they are important not only for understanding biological activity but also for a wide range of diseases.

P22: Lysozyme was purchased from Amresco.

P26:

Data availability

The authors declare that the main data supporting the findings of this study are available within the paper and its associated Supplementary Information. Source data for manuscript figures are provided with this paper.

Code availability

Computer codes developed for amino acid and peptide pattern analyses are provided with the paper.

Further changes in the SI:

P4 The asymmetric error bars for the logarithmic plot were calculated according to eq SI and eq SII for positive σ^+ and negative σ^- errors, respectively.

$$\sigma^+ = \log_{10}(\text{Infection Rel EF-C} + \Delta \text{ Infection Rel EF-C}) - \log_{10} \text{ Infection Rel EF-C} \quad \text{eq SI}$$

$$\sigma^- = \log_{10} \text{ Infection Rel EF-C} - \log_{10}(\text{Infection Rel EF-C} - \Delta \text{ Infection Rel EF-C}) - \quad \text{eq SII}$$

P30 Change mg mL^{-1} to mg/mL

Section 4 add subchapter titles:

4.1 Correlation matrix

4.2 Correlation of infection rates with hydrophobicity and formation of β -sheet rich nanostructures

4.3 Correlation of infection rates and count rate with μm -sized aggregate formation

References for the Point-by-Point Response

- 1 D. Schütz, C. Read, R. Groß, A. Röcker, S. Rode, K. Annamalai, M. Fändrich and J. Münch, *ACS Omega*, 2021, **6**, 7731–7738.
- 2 S. Sieste, T. Mack, E. Lump, M. Hayn, D. Schütz, A. Röcker, C. Meier, K. Kaygisiz, F. Kirchhoff, T. P. J. Knowles, F. S. Ruggeri, C. V. Synatschke, J. Münch and T. Weil, *Adv. Funct. Mater.*, 2021, **31**, 2009382.
- 3 M. Yolamanova, C. Meier, A. K. Shaytan, V. Vas, C. W. Bertoncini, F. Arnold, O. Zirafi, S. M. Usmani, J. A. Müller, D. Sauter, C. Goffinet, D. Palesch, P. Walther, N. R. Roan, H. Geiger, O. Lunov, T. Simmet, J. Bohne, H. Schrezenmeier, K. Schwarz, L. Ständker, W.-G. Forssmann, X. Salvatella, P. G. Khalatur, A. R. Khokhlov, T. P. J. Knowles, T. Weil, F. Kirchhoff and J. Münch, *Nat. Nanotechnol.*, 2013, **8**, 130–136.
- 4 S. Kirti, K. Patel, S. Das, P. Shrimali, S. Samanta, R. Kumar, D. Chatterjee, D. Ghosh, A. Kumar, P. Tayalia and S. K. Maji, *ACS Biomater. Sci. Eng.*, 2019, **5**, 126–138.
- 5 D. Schütz, C. Read, R. Groß, A. Röcker, S. Rode, K. Annamalai, M. Fändrich and J. Münch, *ACS Omega*, 2021, **6**, 7731–7738.
- 6 J. Münch, E. Rücker, L. Ständker, K. Adermann, C. Goffinet, M. Schindler, S. Wildum, R. Chinnadurai, D. Rajan, A. Specht, G. Giménez-Gallego, P. C. Sánchez, D. M. Fowler, A. Koulov, J. W. Kelly, W. Mothes, J. C. Grivel, L. Margolis, O. T. Keppler, W. G. Forssmann and F. Kirchhoff, *Cell*, 2007, **131**, 1059–1071.
- 7 K. Kaygisiz, A. Dutta, L. Rauch-Wirth, C. V Synatschke, J. Münch, T. Bereau and T. Weil, *ChemRxiv*, , DOI:10.26434/chemrxiv-2023-1kqm3.
- 8 D. Schütz, S. Rode, C. Read, J. A. Müller, B. Glocker, K. M. J. Sparrer, O. T. Fackler, P. Walther and J. Münch, *Adv. Funct. Mater.*, 2021, **31**, 2104814.
- 9 B. Schwarze and D. Huster, *Macromol. Biosci.*, 2023, 2200489.
- 10 R. Tycko, *Protein Sci.*, 2014, **23**, 1528–1539.
- 11 F. S. Ruggeri, G. Longo, S. Faggiano, E. Lipiec, A. Pastore and G. Dietler, *Nat. Commun.*, 2015, **6**, 1–9.
- 12 C. Yuan, W. Ji, R. Xing, J. Li, E. Gazit and X. Yan, *Nat. Rev. Chem.*, 2019, **3**, 567–588.
- 13 R. Paparcone, S. W. Cranford and M. J. Buehler, *Nanoscale*, 2011, **3**, 1748.
- 14 J. Adamcik, J.-M. Jung, J. Flakowski, P. De Los Rios, G. Dietler and R. Mezzenga, *Nat. Nanotechnol.*, 2010, **5**, 423–428.
- 15 S. Bolisetty, J. Adamcik and R. Mezzenga, *Soft Matter*, 2011, **7**, 493–499.
- 16 D. M. Ridgley, K. C. Ebanks and J. R. Barone, *Biomacromolecules*, 2011, **12**, 3770–3779.
- 17 T. P. J. Knowles, T. W. Oppenheim, A. K. Buell, D. Y. Chirgadze and M. E. Welland, *Nat. Nanotechnol.*, 2010, **5**, 204–207.

- 18 I. Usov, J. Adamcik and R. Mezzenga, *Faraday Discuss.*, 2013, **166**, 151.
- 19 D. M. Ridgley and J. R. Barone, *ACS Nano*, 2013, **7**, 1006–1015.
- 20 C. Lara, J. Adamcik, S. Jordens and R. Mezzenga, *Biomacromolecules*, 2011, **12**, 1868–1875.
- 21 G. Meisl, L. Rajah, S. A. I. Cohen, M. Pfammatter, A. Šarić, E. Hellstrand, A. K. Buell, A. Aguzzi, S. Linse, M. Vendruscolo, C. M. Dobson and T. P. J. Knowles, *Chem. Sci.*, 2017, **8**, 7087–7097.
- 22 S.-Y. Qin, Y. Pei, X.-J. Liu, R.-X. Zhuo and X.-Z. Zhang, *J. Mater. Chem. B*, 2013, **1**, 668–675.
- 23 A. Sidhu, J. Vaneyck, C. Blum, I. Segers-Nolten and V. Subramaniam, *Amyloid*, 2018, **25**, 189–196.
- 24 E. Arad, H. Green, R. Jelinek and H. Rapaport, *J. Colloid Interface Sci.*, 2020, **573**, 87–95.
- 25 S. Namioka, N. Yoshida, H. Konno and K. Makabe, *Biochemistry*, 2020, **59**, 2782–2787.
- 26 M. Biancalana, K. Makabe, A. Koide and S. Koide, *J. Mol. Biol.*, 2009, **385**, 1052–1063.
- 27 M. Bucciantini, E. Giannoni, F. Chiti, F. Baroni, L. Formigli, J. Zurdo, N. Taddei, G. Ramponi, C. M. Dobson and M. Stefani, *Nature*, 2002, **416**, 507–511.
- 28 M. Stefani and C. M. Dobson, *J. Mol. Med.*, 2003, **81**, 678–699.
- 29 L. Milanese, T. Sheynis, W.-F. Xue, E. V. Orlova, A. L. Hellewell, R. Jelinek, E. W. Hewitt, S. E. Radford and H. R. Saibil, *Proc. Natl. Acad. Sci.*, 2012, **109**, 20455–20460.
- 30 C. Schilling, T. Mack, S. Lickfett, S. Sieste, F. S. Ruggeri, T. Sneideris, A. Dutta, T. Bureau, R. Naraghi, D. Sinske, T. P. J. Knowles, C. V. Synatschke, T. Weil and B. Knöll, *Adv. Funct. Mater.*, 2019, **29**, 1809112.
- 31 K. Kaygisiz, A. M. Ender, J. Gačanin, L. A. Kaczmarek, D. A. Koutsouras, A. N. Nalakath, P. Winterwerber, F. J. Mayer, H. Räder, T. Marszalek, P. W. M. Blom, C. V. Synatschke and T. Weil, *Macromol. Biosci.*, 2023, **23**, 2200294.
- 32 A. M. Ender, K. Kaygisiz, H.-J. Räder, F. J. Mayer, C. V. Synatschke and T. Weil, *ACS Biomater. Sci. Eng.*, 2021, **7**, 4798–4808.

Reviewer #1 (Remarks to the Author):

The authors of the manuscript "Data-Mining Unveils Structure–Property–Activity Correlation of Viral Infectivity Enhancing Self-Assembling Peptides", have answered all of my and other reviewers' queries in a reasonable and logical manner. They have also updated the manuscript to include the answers to the queries and other suggested changes which in my opinion, has improved the quality of the manuscript.

Hence, I recommend the journal to consider the new version of the manuscript for publication.

Reviewer #2 (Remarks to the Author):

The authors have revised their manuscript according to reviewers' comments and the quality of this paper has been enhanced greatly. Therefore, I recommend this paper to be accepted in the present form.

Reviewer #3 (Remarks to the Author):

My comments have been thoughtfully addressed.

Response from the authors

We thank the reviewer for carefully reading our manuscript and the positive evaluation of our first response letter and endorsement of the novelty and relevance of our work.

Reviewer #1 (Remarks to the Author):

The authors of the manuscript "Data-Mining Unveils Structure–Property–Activity Correlation of Viral Infectivity Enhancing Self-Assembling Peptides", have answered all of my and other reviewers' queries in a reasonable and logical manner. They have also updated the manuscript to include the answers to the queries and other suggested changes which in my opinion, has improved the quality of the manuscript.

Hence, I recommend the journal to consider the new version of the manuscript for publication.

Reviewer #2 (Remarks to the Author):

The authors have revised their manuscript according to reviewers' comments and the quality of this paper has been enhanced greatly. Therefore, I recommend this paper to be accepted in the present form.

Reviewer #3 (Remarks to the Author):

My comments have been thoughtfully addressed.